# Selective silencing of antibiotic-tethered ribosomes as a resistance mechanism against aminoglycosides

Nilanjan Ghosh Dastidar [1,2,8], Nicola S. Freyer [1,2,8], Valentyn Petrychenko [3], Ana C. de A. P. Schwarzer [3], Bee-Zen Peng [1], Ekaterina Samatova[1], Christina Kothe[1], Marlen Schmidt[4], Frank Peske[1], Antonio Z. Politi[5], Henning Urlaub [6,7], Niels Fischer [3], Marina V. Rodnina [1] ✉ & Ingo Wohlgemuth [2] ✉

Antibiotic resistance is a growing threat, underscoring the need to understand the underlying mechanisms. Aminoglycosides kill bacteria by disrupting translation fidelity, leading to the synthesis of aberrant proteins. Surprisingly, mutations in *fusA*, a gene encoding translation elongation factor G (EF-G), frequently confer resistance, even though EF-G neither participates in mRNA decoding nor blocks aminoglycoside binding. Here, we show that EF-G resistance variants selectively slow ribosome movement along mRNA when aminoglycosides are bound. This delay increases the chance that the drug dissociates before misreading occurs. Over several elongation cycles, this selective silencing of drug-bound ribosomes prevents error cluster formation, preserving proteome and membrane integrity. As a result, *fusA* mutations confer resistance early in treatment by preventing self-promoted aminoglycoside uptake. Translation on drug-free ribosomes remains sufficiently rapid to sustain near-normal bacterial growth. The mechanism of selective silencing of corrupted targets reveals a previously unrecognized antibiotic resistance strategy with potential therapeutic implications.

Infections with antibiotic-resistant bacteria pose an increasing threat to human health, ranking globally as the third leading cause of death in 2019[1]. Most aminoglycoside antibiotics (AGAs) are bactericidal, broad-spectrum antimicrobials that reduce the speed and fidelity of translation (Supplementary Fig. 1a–c)[2–4]. In Gram-negative bacteria, AGAs must cross both the outer and inner membranes to reach the ribosome, with the inner membrane representing a major permeability barrier for these cationic, hydrophilic compounds. According to the current uptake model[5–7], AGAs enter the periplasm through porin channels or by disrupting the outer membrane. Aided by the membrane potential, they then seep into the cytoplasm, where they bind to a few ribosomes and induce mRNA misreading[2,8,9] by stabilizing an error-prone ribosome conformation[10]. Even single amino acid substitutions can render proteins nonfunctional; however, when AGA-bound ribosomes continue translation in the error-prone conformation[3,11], they can accumulate consecutive amino acid

[1]Department for Physical Biochemistry, Max Planck Institute for Multidisciplinary Sciences, Göttingen, Germany. [2]Project Group Fidelity of Protein Synthesis in vivo, Department for Physical Biochemistry, Max Planck Institute for Multidisciplinary Sciences, Göttingen, Germany. [3]Project Group Molecular Machines in Motion, Department for Physical Biochemistry, Max Planck Institute for Multidisciplinary Sciences, Göttingen, Germany. [4]Genetic Engineering Heidelberg GmbH, Heidelberg, Germany. [5]Facility for Light Microscopy, Max Planck Institute for Multidisciplinary Sciences, Göttingen, Germany. [6]Bioanalytical Mass Spectrometry Group, Max Planck Institute for Multidisciplinary Sciences, Göttingen, Germany. [7]Institute of Clinical Chemistry, Bioanalytics, University Medical Center Göttingen, Göttingen, Germany. [8]These authors contributed equally: Nilanjan Ghosh Dastidar, Nicola S. Freyer. ✉e-mail: rodnina@mpinat.mpg.de; Ingo.Wohlgemuth@mpinat.mpg.de

substitutions known as error clusters[12], further compounding the loss of protein function. Error clusters potentiate proteotoxic error burden at the onset of AGA treatment, when only a small fraction of ribosomes is corrupted by the drug. The accumulation of faulty membrane proteins[13–15] and misreading-induced metabolic by-products[16,17] makes the inner membrane permeable to AGAs. As more and more AGAs enter the cell, more ribosomes bind AGAs and become corrupted. Over time, this self-promoted uptake leads to a massive influx of AGAs, resulting in a burst of translation errors, proteostasis collapse, and an accumulation of reactive by-products. These factors damage macro-molecular structures[16,18], disrupt metabolism[19], and cause membrane voltage dysregulation[20], ultimately leading to cell death.

In general, AGA resistance can be achieved through intrinsic, adaptive, or acquired mechanisms[21]. Gram-negative bacteria, such as *Escherichia coli*, are naturally protected against rapid AGA uptake due to their membrane architecture and efflux pumps, such as AcrAD[22]. Bacteria can also develop tolerance by adopting a different lifestyle as persisters or within biofilms[23]. Acquired resistance is typically medi-ated by drug-modifying enzymes or ribosomal methyltransferases spread by horizontal gene transfer. While these enzymes provide high-

level resistance, they impose a metabolic and genetic burden and are specific to a narrow range of drugs[21,24]. AGA resistance can also arise as a result of mutations in the translational apparatus. Although 16S rRNA mutations in the ribosomal decoding center can confer high-level resistance[25], their impact is limited because most pathogens have multiple rRNA-encoding genes, so acquiring mutations in all copies simultaneously is unlikely. AGA resistance mutations can also arise in ribosomal proteins (Supplementary Fig. 1a); however, most AGA resistance mutations occur outside of the ribosome itself.

Unexpectedly, a major hotspot of AGA resistance mutations is the *fusA* gene, which encodes elongation factor G (EF-G), a GTPase that promotes translocation of mRNA and tRNAs on the ribosome during each elongation cycle of protein synthesis (Supplementary Fig. 1d and Supplementary Data 1). This is surprising, because EF-G is not directly involved in decoding and therefore cannot counteract AGA-induced misreading. Additionally, EF-G does not interact with ribosome-bound AGA, so it cannot directly displace AGA from its binding site (Fig. 1a). Nevertheless, *fusA* mutations confer resistance in various bacteria (Supplementary Data 1) as well as in parasites such as *Leishmania*[26]. For instance, clinical isolates of *Pseudomonas aeruginosa* lacking

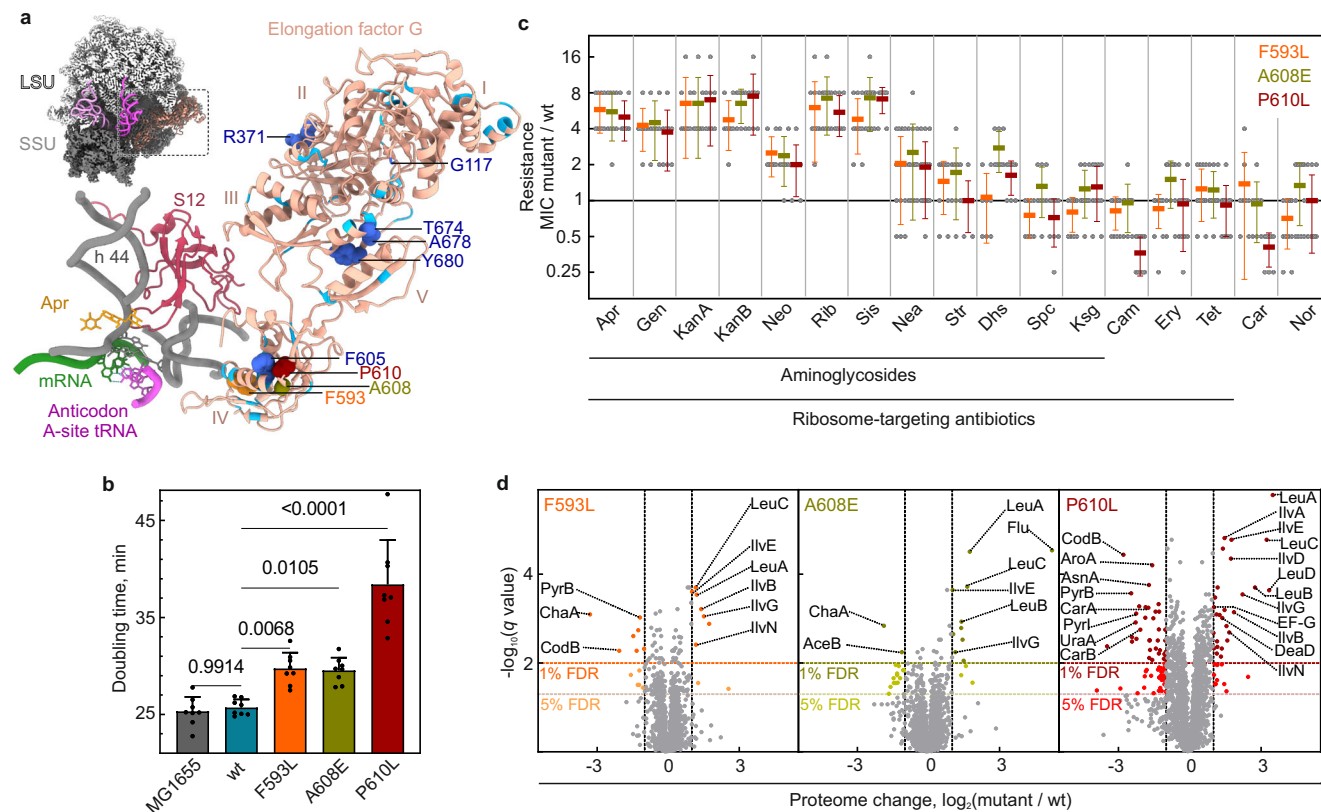

**Fig. 1 | *fusA* mutations confer resistance to a specific set of AGAs. a** Resistance mutations in EF-G domains I to V. Mutations are mapped onto the structure of EF-G bound to the ribosome. An overview of the close-up is shown in the upper left for orientation. Close-up: AGA Apr and the key structural elements at the decoding center are shown (PDB entry 7PJV)[51]. Mutations that evolve during AGA treatment in *E. coli* are shown in cyan (Supplementary Data 1). Resistance variants F593L (orange), A608E (olive), and P610L (red) were chosen for in-vivo experiments using chromosomally-encoded *fusA* mutations. These and several other mutations (dark blue) were also introduced into the plasmid-encoded *fusA* gene for expression of EF-G variants, which were then used in translocation experiments in vitro. Notably, none of the residues at the mutated positions is close to the ribosome-bound Apr (> 20 Å in all cases) and none is involved in direct interactions with the ribosome. **b** Effect of chromosomally-encoded *fusA* mutations on cell growth. Shown are mean doubling times ± SD of 8 biological replicates (*n* = 8). Significance levels were

determined by one-way ANOVA followed by Dunnett's test (two-sided); adjusted *P* values are indicated in the figure. **c** Spectrum of antibiotic resistance. Minimal inhibitory concentrations (MIC) were measured by broth microdilution. Horizontal lines represent the MIC ratio of *fusA* mutant and wt strain. Mean values ± SD of ≥8 biological replicates are shown (exact *n* is specified in the Source Data file). **d** Volcano plots representing log₂-fold changes in protein abundance (X axis) and negative log₁₀ adjusted (Benjamini-Hochberg) *P* values (Y axis) for *fusA* mutants F593L (left panel), A608E (center panel), and P610L (right panel) compared to the wt (*n* = 4 biological replicates). Colored dots represent proteins which are sig-nificantly (*q* value < 0.01, more-saturated colors; *q* value < 0.05, less-saturated col-ors) and more than 2-fold regulated. Horizontal dashed lines indicate 1% and 5% false discovery rate (FDR) thresholds; vertical dashed lines indicate a two-fold change threshold. Source data are provided as a Source Data file.

AGA-inactivating enzymes often harbor *fusA1* mutations[27]. In adaptive laboratory evolution experiments, *fusA* mutations rapidly evolve in various pathogens, including *Mycobacterium tuberculosis*[28] and all ESKAPE pathogens (*Enterococcus faecium*, *Staphylococcus aureus*, *Klebsiella pneumoniae*, *Acinetobacter baumannii*, *Pseudomonas aeruginosa*, and *Enterobacter cloacae*)[29,30], which are responsible for most multidrug-resistant infections. In *E. coli*, *fusA* mutations emerge under selection by various AGAs such as amikacin (Amk), apramycin (Apr), gentamicin (Gen), kanamycin A (KanA), and tobramycin (Tob) (Supplementary Fig. 1d)[30–41], but not with AGAs like neamine (Nea), streptomycin (Str), or spectinomycin (Spc) (Supplementary Fig. 1a)[40]. In many cases, *fusA* mutations alone are sufficient to confer AGA resistance[37,42,43], with most mutations imposing minimal to moderate fitness costs[37,39], and complementing these strains with wild type (wt) EF-G restores AGA sensitivity[37,44].

Despite their clinical relevance, the mechanism by which *fusA* mutations confer resistance remains unclear, as EF-G cannot directly block AGA binding or contact the decoding center (Fig. 1a). In this study, we used a combination of quantitative mass spectrometry (MS), live-cell imaging, kinetic analysis, and cryo-EM to identify a novel type of resistance mechanism by which *fusA* mutations silence corrupted ribosomes and shield cells from AGAs.

## Results

### Effect of *fusA* mutations on bacterial growth, antibiotic resistance, and proteome composition

To explore how *fusA* mutations confer resistance to AGAs, we constructed *E. coli* MG1655 strains harboring three frequently reported laboratory-evolved EF-G variants (F593L, A608E, and P610L) (Fig. 1a), which belong to a prominent cluster of resistance mutations in EF-G domain IV (Supplementary Data 1), along with an isogenic wild-type control generated by the same cloning strategy but retaining the wt sequence (see "Methods", Supplementary Fig. 3a-d). Resistance mutations in the same region of EF-G have been identified in ESKAPE pathogens, suggesting a shared resistance mechanism against AGAs across different bacterial species[29]. EF-G variants F593L and A608E caused only a slight reduction in the growth rate, whereas P610L had a stronger effect (Fig. 1b). All three mutations conferred resistance to a range of AGAs, including Apr, Gen, KanA, KanB, neomycin (Neo), ribostamycin (Rib), and sisomicin (Sis) (Fig. 1c). Despite belonging to different structural classes of AGAs (Supplementary Fig. 1a), these AGAs share a common mode of action: they slow down translocation and induce misreading and error cluster formation[12]. The P610L mutant has a stronger negative impact on cell growth (Fig. 1b), yet it conferred the same level of resistance as the F593L and A608E mutants. This suggests that the reduced growth rate alone cannot fully explain the observed resistance, thus challenging models that attribute resistance to just a slower translation rate, which would provide the cell with more time to pump out AGAs (see Supplementary Fig. 2 for details of putative resistance mechanisms; models 1 and 2). Furthermore, the mutations did not confer significant resistance to neamine (Nea), which slows translocation and induces misreading at high concentrations, but does not induce error clusters[12]. The mutant strains also remained sensitive to streptomycin (Str), which induces errors and error clusters[12] but does not strongly inhibit translocation[45]. These observations show that the mutant strains are not generally tolerant towards translational misreading and proteotoxic stress. Sensitivity to spectinomycin (Spc), which inhibits translocation without inducing misreading, and kasugamycin (Ksg), which does not bind to the ribosomal A site, remained unchanged, further supporting the notion that *fusA* mutant strains do not exhibit broad resistance to all AGAs. Together, these observations suggest that *fusA* mutations specifically counteract the effects of AGAs that both disrupt translocation and induce misreading with error cluster formation.

Furthermore, the *fusA* mutations did not confer resistance to other bactericidal antibiotics such as carbenicillin (Car), a cell wall synthesis inhibitor, or norfloxacin (Nor), a gyrase inhibitor. Since these antibiotics—as well as some AGAs—have been reported to kill bacteria by escalating metabolic stress[14,46], the lack of resistance suggests that *fusA* mutations do not mitigate this common metabolic death pathway. Instead, their resistance appears to be specific to a subset of AGAs, likely through a more targeted mechanism affecting translation.

To determine whether *fusA* mutations might confer resistance to AGAs indirectly by inducing expression of resistance proteins or redirecting existing pathways (Supplementary Fig. 2, model 2), we analyzed the proteomes of the three mutant strains compared to the parental MG1655 and wt strains. The *fusA* mutations did not alter the expression patterns of proteins directly associated with AGA entry (e.g., porins, sugar and peptide transporters), AGA removal (e.g., efflux pumps), AGA-induced stress adaptation (e.g., ribosome silencing factors, global stress modulators), AGA response regulation (e.g., stress-associated transcription factors), and AGA-induced proteostasis maintenance (e.g., proteases and chaperones) (Supplementary Fig. 4a). Overall, the proteomic changes induced by the mutations were minimal for F593L and A608E, and moderate for P610L, with at most 63 out of 2527 proteins showing more than a 2-fold change (Fig. 1d), and correlated with growth rate differences (Fig. 1b and Supplementary Fig. 4b–d) but not with resistance patterns (Fig. 1c and Supplementary Fig. 4e). Thus, these adaptations are likely due to changes in translation rates (Fig. 1d and Supplementary Fig. 4b–f), but do not account for the observed AGA resistance, further challenging the validity of models suggesting an indirect effect of *fusA* mutations on adaptation (Supplementary Fig. 2, model 2).

Notably, the most upregulated proteins, such as enzymes of the Leu and Ile synthesis pathway (Fig. 1d), are regulated by translation attenuation mechanisms normally activated during starvation. Under starvation, translation slows down due to low aminoacyl-tRNA levels, promoting the formation of alternative mRNA secondary structures that enhance full-length transcription of Leu and Ile biosynthetic operons, resulting in increased expression levels of enzymes involved in Leu and Ile synthesis[47]. The same regulatory pathway is activated in *fusA* mutant strains, presumably due to slower translation, stimulating the production of these anabolic enzymes under non-starvation conditions. A similar misregulation seems to occur in the pyrimidine synthesis pathway in *fusA* mutant strains. Under nutrient-rich conditions, high UTP levels speed up transcription of uridine-rich regulatory sequences, resulting in the loss of transcription-translation coupling and downregulation of key enzymes of pyrimidine biosynthesis, PyrB and PyrI. The observed down regulation of PyrB and PyrI in *fusA* mutants (Fig. 1d) is consistent with previous reports on the effects of ribosome variants[48], which showed that slower translation leads to a transcription-translation decoupling and a misregulation of the expression of the *pyrBI* operon. While *fusA* mutations activate attenuation pathways, suggesting alterations in local translation speed, they have little effect on cell growth, implying that EF-G variants still promote efficient mRNA translocation.

### EF-G mutations can selectively slow down translation on AGA-bound ribosomes

To test how *fusA* mutations affect translation, we performed in-vitro experiments with EF-G variants F593L, A608E, and P610L, as well as additional variants within the prominent mutation cluster in domain IV (F593C, F605I, A608V, and P610Q). We also tested five additional mutations (G117C, R371L, T674A, A678V, and Y680C) located throughout different domains of EF-G (Fig. 1a and Supplementary Data 1). Notably, the last three mutations (T674A, A678V, and Y680C) belong to a cluster of mutations that are frequently found in *P. aeruginosa* (residues 668-680). We analyzed the in-vitro translocation activity of the purified EF-G variants in the absence of AGAs using a

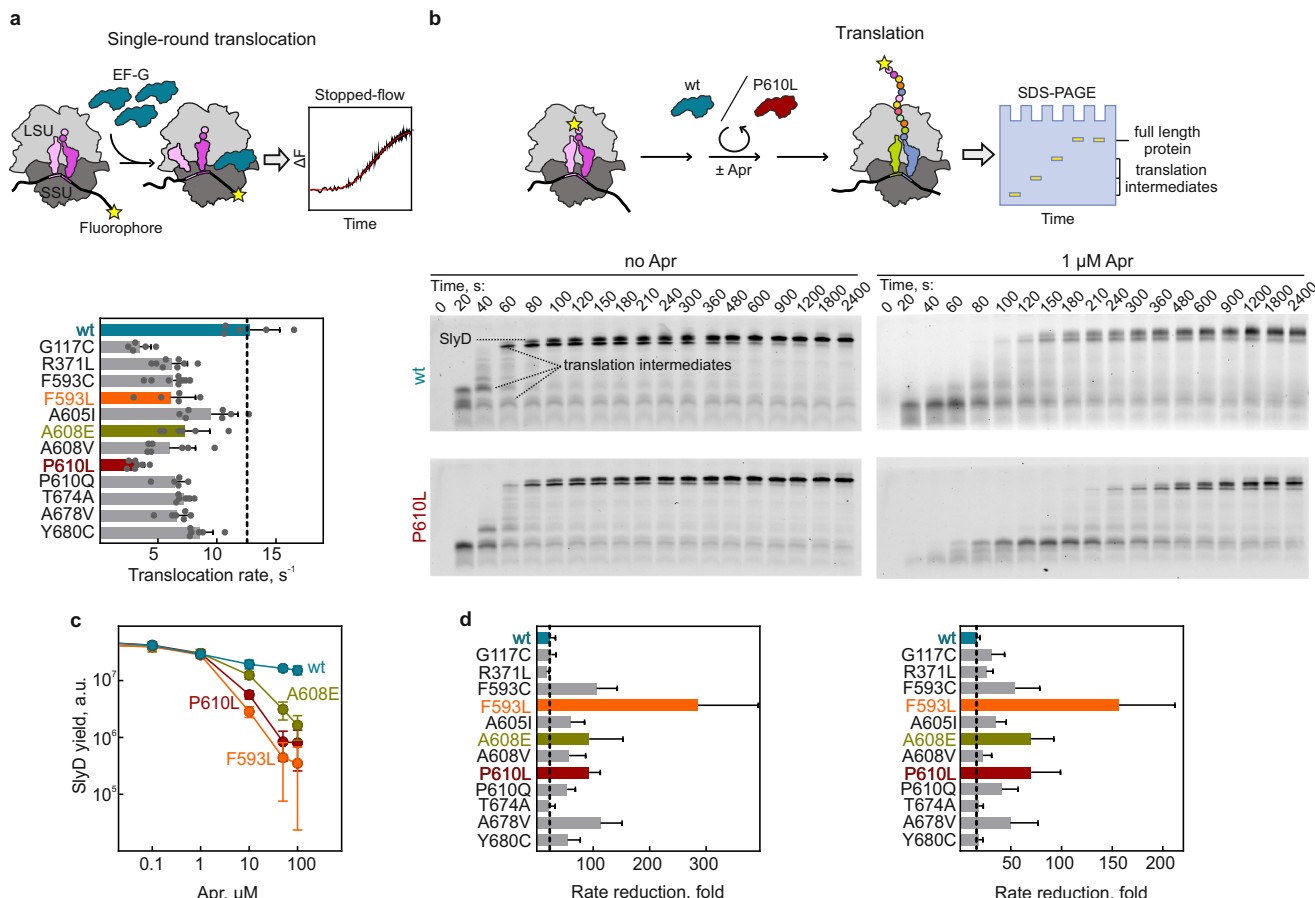

**Fig. 2 | EF-G variants selectively slow down AGA-bound ribosomes.**
**a** Translocation rates in the absence of AGAs. Top panel: Schematics of transloca-tion assay. Ribosome movement by one codon was measured under single round translocation conditions on synchronized ribosomes in excess of EF-G by tracking the fluorescence change of fluorescein-labeled model mRNA in a stopped-flow apparatus. SSU, small subunit; LSU, large subunit. Bottom panel: Bar graph showing translocation rates calculated by exponential fitting of the stopped-flow time courses shown as mean values ± SD of ≥5 technical replicates (exact $n$ per bar are specified in the Source Data file). Dotted line indicates the wt rate. **b** In-vitro translation in the absence and presence of AGA. Top panel: Schematics of trans-lation assay. Ribosome initiation complexes with *slyD* mRNA and BodipyFL-labeled Met-tRNA$^{fMet}$ were mixed with EF-G (P610L or wt), and ternary complexes of

aminoacyl-tRNAs, GTP, and elongation factor Tu (EF-Tu, not shown). The reaction was stopped after indicated time intervals and translation products were separated by Tris-Tricine SDS-PAGE and visualized using the N-terminal BodipyFL fluores-cence reporter. Bottom panel: Time courses of *slyD* translation with wt EF-G and P610L in the absence and presence of Apr (1 μM). **c** Translation yield of full-length SlyD at increasing Apr concentrations after 30 min of in-vitro translation. Mean values ± SD of 3 biological replicates are shown ($n$ = 3). **d** Effect of AGAs on single-round translocation rates at saturating concentrations of Apr (left panel) or KanA (right panel) (both 250 μM). Bars represent mean values ± SD of ≥3 technical replicates (underlying values are provided in the Source Data file). Dotted line indicates the AGA-induced rate reduction for EF-G wt. Source data are provided as a Source Data file.

time-resolved assay that monitors the kinetics of individual transloca-tion events[49]. The EF-G variants exhibited a moderate reduction in translocation rates, ranging from 1.3- to 4-fold (Fig. 2a). For the F593L, A608E, and P610L variants, the results align with the observed changes in growth rates and the proteome changes (Supplementary Fig. 5a). Similarly, a translation assay monitoring the time course of in-vitro synthesis of the model protein SlyD revealed that EF-G variants have only minor effects on translation speed in the absence of AGA (Fig. 2b and Supplementary Fig. 5b).

In the presence of low, subsaturating Apr concentrations, translation with wt EF-G was slower, as expected[12,50]. Notably, the EF-G resistance variants did not alleviate the inhibitory effect of Apr; instead, translation was even slower than with wt EF-G. This suggests that EF-G resistance variants neither displace the drug from the ribosome nor enable efficient translation on AGA-bound ribosomes, ruling out the 'drug displacement' and 'gain-of-function' models proposing that variant EF-G accelerates translocation in the pre-sence of the drug (Supplementary Fig. 2, models 3,4). With increasing Apr concentration, more ribosomes bound Apr, leading to a dramatic decrease in the yield of full-length SlyD (Fig. 2c and

Supplementary Fig. 5c). While translation with wt EF-G continued to produce SlyD even at high Apr concentrations, the EF-G resistance variants substantially reduced SlyD formation at concentrations above 10 μM, indicating that these variants specifically impair translation on the AGA-bound ribosomes. To better assess the underlying kinetic effect on translocation on AGA-bound ribosomes, we measured the rates of individual translocation events at high Apr and KanA concentrations (250 μM), at which nearly all ribosomes carry AGAs (Fig. 2d). As expected, Apr and KanA substantially reduced translocation rates for wt EF-G, without completely block-ing it. Notably, none of the twelve EF-G variants alleviated the translocation inhibition (Fig. 2d), effectively ruling out resistance mechanisms in which EF-G variants compensate for the inhibitory AGA effect (Supplementary Fig. 2, model 3) or displace the drug allosterically (Supplementary Fig. 2, model 4). Instead, for most EF-G variants, especially the frequently observed variants in EF-G domain IV including F593L, A608E, and P610L, translocation rates sub-stantially decreased in the presence of AGAs, supporting the idea that these mutants are selectively hindered in promoting translo-cation on AGA-bound ribosomes (Supplementary Fig. 2, model 5).

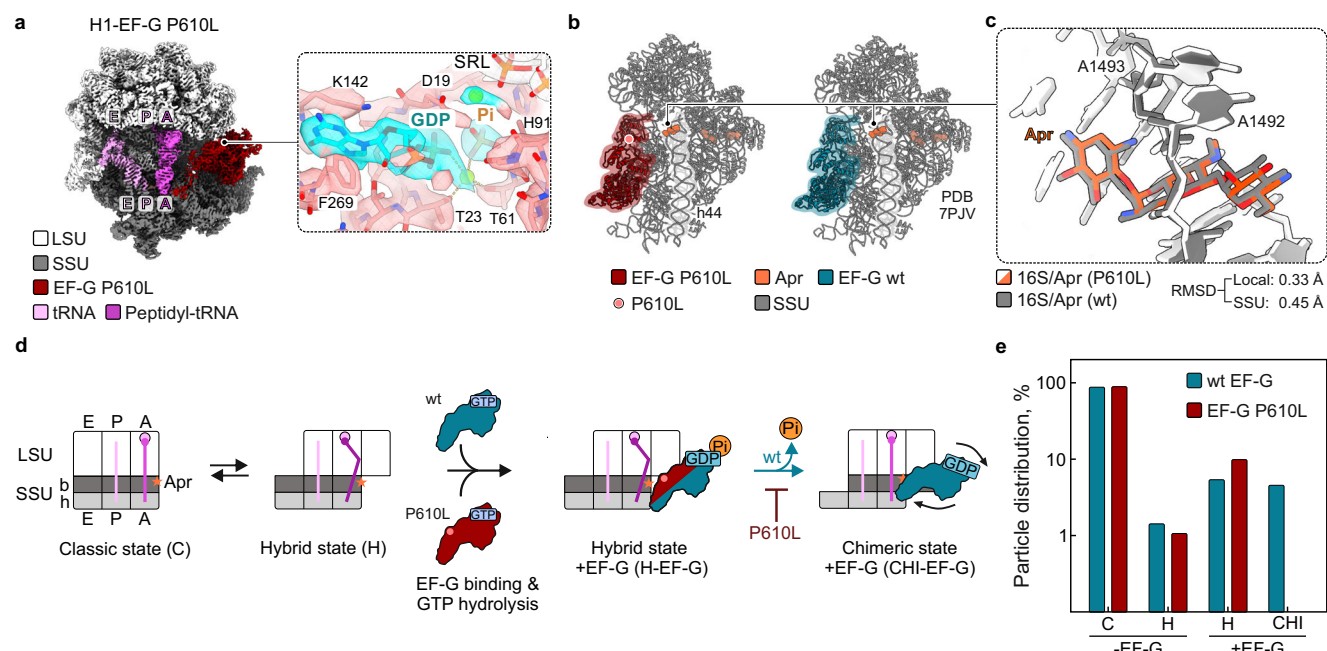

**Fig. 3 | EF-G variant P610L selectively impedes translation on Apr-bound ribosomes. a** Cryo-EM map of EF-G P610L bound to 70S ribosome in the presence of Apr at 3.0 Å resolution. SSU (dark gray) and LSU (light gray), small and large ribosomal subunit, respectively. A, P, E are tRNA binding sites on SSU and LSU with the two tRNAs (deacylated tRNA in the P site (pink) and peptidyl-tRNA in the A site (magenta) in hybrid states and EF-G in the GDP-Pi form. Close-up: Nucleotide binding pocket of EF-G P610L with GDP-Pi; residues of EF-G taking part in stabilization of GDP-Pi, including H91 which is a key residue for GTP hydrolysis, are indicated. SRL, sarcin-ricin loop of LSU. **b** Comparison of the SSU conformations in complexes with EF-G P610L-GDP-Pi (left) and wt EF-G-GDP-Pi (right)[51]. The mRNA and tRNAs were omitted for visual clarity. Pink dot, location of P610L substitution in EF-G. **c** Superposition of Apr-binding site from structures shown in (**b**), demonstrating that EF-G wt and P610L do not displace the antibiotic. A1492 and A1493 are key 16S rRNA residues in the decoding center[4,10]; RMSD, root-mean-square

deviation. **d** Mechanism of translocation inhibition. Schematic of translocation in the presence of Apr with EF-G wt (blue) or P610L (red). b and h, body and head of the SSU, respectively. The ribosome-tRNA complexes are shown in three conformations: classical C state; hybrid H state with the SSU rotated relative to LSU; and chimeric CHI state with the head and body domains of the SSU swiveled relative to each other and tRNAs partially translocated relative to the SSU body domain. Both EF-G wt and P610L bind to the ribosome[51], stabilize tRNA hybrid states and hydrolyze GTP. Upon Pi release, wt EF-G promotes tRNA movement relatively to the SSU resulting in the formation of the CHI state, a bona-fide translocation intermediate. In contrast, in EF-G P610L, the key reaction of Pi release is slowed down, inhibiting the conformational rearrangement required to promote the tRNA movement into the CHI state. **e** Cryo-EM particle distribution. In contrast to wt EF-G[51], EF-G P610L impedes transition to chimeric (CHI) tRNA states in presence of Apr. Source data are provided as a Source Data file.

Next, we used time-resolved cryo-EM to visualize the mechanism of translation inhibition and confirm that the AGA is not displaced. To capture various states of translocation, we slowed the reaction by lowering the temperature and by adding polyamines[51]. Our cryo-EM analysis of EF-G P610L-promoted translocation reveals that the EF-G variant and Apr bind simultaneously to the ribosome (Fig. 3a–c, and Supplementary Fig. 6a–f and Supplementary Table 1). Superimposing the Apr-bound ribosome-EF-G P610L complex with the previously published Apr-bound ribosome EF-G wt complex[51] shows no significant structural changes in the decoding center or drug-binding site (Fig. 3c). These findings further support the notion that EF-G variants do not displace the drug from the ribosome (Supplementary Fig. 2, model 3).

With wt EF-G, cryo-EM structures and particle populations reveal that Apr does not interfere with the early steps of translocation, including tRNA hybrid state stabilization upon EF-G binding, GTP hydrolysis, inorganic phosphate (Pi) release, up to the movement of tRNAs into chimeric states on the small ribosomal subunit (SSU) (Fig. 3d,e)[51]. However, Apr significantly slows down the final transition of the tRNAs on the SSU to the post-translocation state[51]. With EF-G P610L, we could not detect any particles in the chimeric state, despite extensive sorting of the cryo-EM data (see "Methods"). Furthermore, we could also not find any post-Pi-release state that would correspond to a tRNA hybrid state with EF-G-GDP bound, as we found previously for wt EF-G[51]. Accordingly, with EF-G P610L the early stages of translocation up to GTP hydrolysis are unaffected by Apr but the later steps appear to be inhibited, particularly Pi release and the tRNA movement

into chimeric states. The fact that EF-G P610L and Apr can bind simultaneously, and that the mutation impairs late translocation steps, is inconsistent with a 'gain-of-function' model (Supplementary Fig. 2, model 4). Together with the kinetic analysis (Fig. 2), these results demonstrate that EF-G variants are impaired in facilitating translation on AGA-bound ribosomes. Furthermore, this also suggests that EF-G resistance variants can promote efficient translation only when intracellular AGA concentrations are low and most ribosomes are drug-free, whereas at higher AGA concentrations, translation on all ribosomes would cease, abolishing the resistance phenotype.

### *fusA* mutations help to preserve proteome integrity

Next, we asked whether selective silencing of AGA-bound ribosomes by EF-G variants affects proteome integrity. To measure translation errors, we treated *E. coli* cultures (wt, F593L, A608E, and P610L) with Apr (Fig. 4a) and quantified amino acid misincorporation using MS. After Apr addition, the cultures continued to grow at similar rates initially, likely due to the lag phase in self-promoted AGA uptake (Fig. 4a). After 60 min, wt cells stopped growing, while mutant cells remained unaffected by the Apr exposure, consistent with their resistance phenotypes (Fig. 1c). In wt cells, the Apr treatment led to a burst of misreading within 60 min (Fig. 4b and Supplementary Fig. 7a–c). After that, error frequencies did not increase further, likely because proteostasis collapsed and cell growth ceased. In contrast, error levels in mutant strains increased gradually and remained lower than in wt cells, suggesting that *fusA* mutations establish resistance early in the initial phase of AGA uptake. As the accumulation of aberrant membrane proteins is crucial for AGA

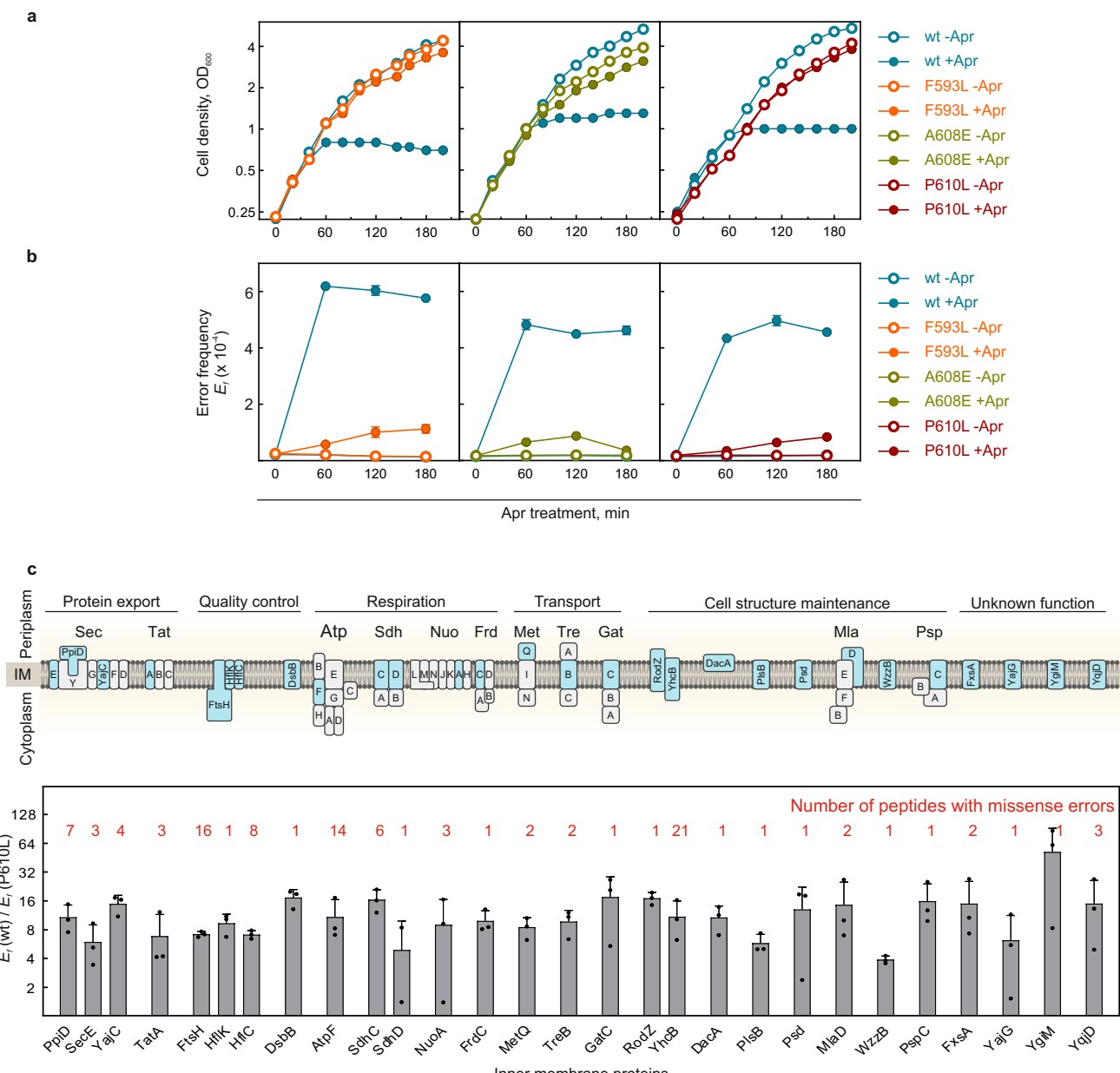

**Fig. 4 | *fusA* mutations preserve proteome integrity. a**, **b** *fusA* mutations preserve cell growth (**a**) and prevent translational misreading (**b**) in the presence of Apr. Cells were treated with Apr (16 μM) at 0.25 OD₆₀₀ and cell growth was recorded photometrically. Peptides with missense errors in the abundant cytosolic protein EF-Tu were quantified by targeted MS using label-free Parallel Reaction Monitoring (PRM). Dots represent the mean ± SD of three technical replicates (*n* = 3) of median error frequencies of at least 25 independent misreading events. Note that in (**b**), untreated wt and mutants are indistinguishable. **c** *fusA* mutations preserve the inner membrane proteome integrity. Inverted inner membrane vesicles were prepared from untreated and Apr-treated (16 μM, 60 min) wt and P610L cells and error frequencies in inner membrane proteins (indicated in the cartoon) were quantified by targeted MS (label-free PRM). Bars represent the relative difference of error frequencies in wt and P610L cells after Apr treatment. Values represent the mean ± SD of 3 biological replicates (*n* = 3) based on 1-21 missense peptides/protein (as indicated). Source data are provided as a Source Data file.

uptake[5], we analyzed error frequencies specifically in membrane proteins (see "Methods"). Similar to their effect on cytosolic proteins, *fusA* mutations helped to maintain the integrity of the membrane proteome, including diverse proteins involved in protein export, quality control, respiration, metabolic transport, and cell structure maintenance (Fig. 4c and Supplementary Fig. 7d).

One type of errors that crucially depends on the translation velocity is error clusters: Faster translation in wt strain should allow AGA-bound ribosomes to decode multiple codons before the drug dissociates, thereby increasing the probability of long error clusters. In contrast, selective silencing of AGA-bound ribosomes in *fusA* mutant strains increases the chance that AGA will dissociate from the ribosome before several elongation cycles are completed, thereby suppressing error cluster formation. To assess whether *fusA* mutations decelerate translation strongly enough for AGA to dissociate before multiple elongation cycles are completed, we measured error cluster formation in wt and mutant strains in vivo using targeted MS (see "Methods")[12]. In wt cells, Apr treatment led to a rapid increase in single translation errors and error clusters and to growth arrest (Fig. 5a, b and Supplementary Fig. 8a). In contrast, in the P610L strain, the level of single errors increased gradually, reaching levels comparable to the wt only at very high Apr concentrations (128-256 μM), reflecting the attenuated AGA uptake at lower concentrations. Across all concentrations and incubation times, error clusters were dramatically

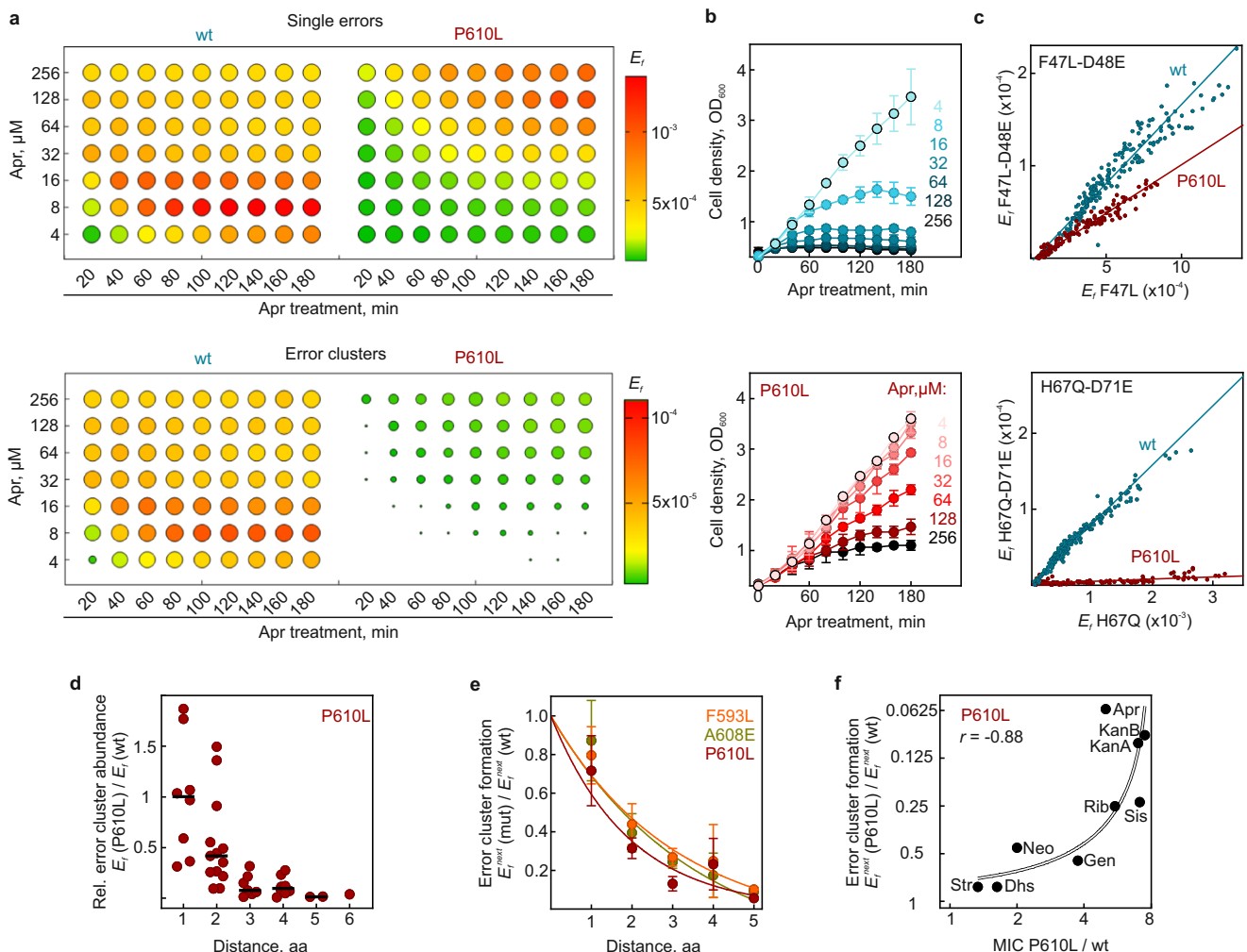

**Fig. 5 | *fusA* mutations substantially reduce error cluster formation.**
**a** Misreading errors and error clusters in Apr-treated wt and P610L cells. Cells were treated at 0.25 OD$_{600}$ with increasing concentrations of Apr (4-256 μM), and misreading for single errors (examined in 12 peptides) and error cluster formation (double errors in 14 peptides) were quantified at various incubation times by targeted MS. The mean of their median error frequencies from 3 biological replicates are color coded (*n* = 3). At high error burden all missense peptides were confidently detected in wt cells (ratio dot products > 0.97, see Supplementary Fig. 8a). To provide an orthogonal measure of misreading across all conditions, the number of detected missense peptides (ratio dot products > 0.8) for each conditions is represented by the size of the circles, with smaller dots reflecting fewer detected peptides. The apparently lower error frequency at Apr concentrations ≥32 μM is due to massive cell death and the rapid decline in actively growing cells. **b** Growth curves of Apr-treated wt and P610L cells used to evaluate translation errors in (**a**). Cell growth was recorded photometrically. Shown are mean values ± SD of 3 biological replicates (*n* = 3). **c** Examples of correlations between error frequencies (*E$_f$*) of the initial misreading event and of corresponding error clusters across 189 biological conditions (from (**a**)). Notably, the ratio depends on the error pair, but is independent of the incubation time and Apr concentration. **d** Distance dependence of error cluster frequencies in P610L vs. wt. Dots represent the mean of three technical replicates (*n* = 3) of individual error clusters compared at conditions of similar error load (wt at 16 μM Apr; P610L at 256 μM), with their median indicated as horizontal line. The distribution of single-error peptides is shown in Supplementary Fig. 8b for comparison. **e** Distance dependence of error cluster formation (*E$_f^{next}$*) in F593L, A608E, and P610L strains normalized to the wt. Dots represent mean values ± SD of 1-4 error clusters per distance, each based on three technical replicates (*n* = 3). **f** Correlation between the reduction in error cluster formation and increase in MIC values for different AGAs (from Fig. 1c) for P610L vs. wt strain (Pearson coefficient *r* = −0.88, *p* = 0.017, two-sided test). Points represent the mean *E$_f^{next}$* values of 5 error clusters with a distance of 2-4 residues between misincorporated amino acids, measured at conditions of induced single errors in wt and P610L (wt: at 16 μM Str, 16 μM Apr, 16 μM Neo, 8 μM Dhs, 4 μM Gen, 32 μM Rib, 16 μM Sis, 8 μM KanA, and 4 μM KanB; P610L: at 16 μM Str, 256 μM Apr, 128 μM Neo, 8 μM Dhs, 32 μM Gen, 256 μM Rib, 64 μM Sis, 256 μM KanA, and 64 μM KanB). The curve represents a linear regression as visual guide; note the logarithmic scale. Source data are provided as a Source Data file.

reduced, with some falling below the mass spectrometer's detection limit.

Error cluster formation depends linearly on the occurrence of the initial error[12] (Fig. 5c), with the slope reflecting the probability that an AGA-bound ribosome, after making the first error, will proceed to make a subsequent error[12]. For example, for the wt strain, the probability of misreading the next codon D48E after the initial F47L misincorporation (*E$_f^{next}$*) is ~0.2 (e.g., 20%), which is a very high error rate (Fig. 5c, upper panel). Shallower slopes indicate that P610L systematically prevents error cluster formation, e.g., for the same error

cluster, *E$_f^{next}$* is ~0.1 − about 2-fold less than in the wt strain. However, the reduction is not uniform for all error clusters (compare examples of F47L-D51E and H67Q-D71E in Fig. 5c). Notably, the codon distance between the consecutive misreading events matters: for the clusters where the misreading events are separated by 3 codons, the probability to make a second error is decreased about 30-fold, from *E$_f^{next}$* ~0.08 to ~0.003. A systematic analysis for P610L showed that clusters with a longer codon distance decrease dramatically compared to the wt (Fig. 5d and Supplementary Fig. 8a, b), while misreading events involving single amino acid substitutions are less affected

(Supplementary Fig. 8b). Similar length-dependent reductions in error cluster formation were also observed for F593L and A608E mutant strains (Fig. 5e), indicating that translocation inhibition of AGA-bound ribosomes and the corresponding reduction in error clusters are hallmarks of *fusA* mediated resistance.

To evaluate whether silencing of AGA-bound ribosomes is associated with resistance, we correlated the effect of different AGAs on P610L-mediated error cluster suppression with their impact on resistance (Fig. 5f). For Apr, KanA, KanB, Sis, and Rib, error clusters were strongly reduced in P610L compared to wt, correlating with a significant increase in resistance. In contrast, Neo and Gen showed a smaller reduction in error cluster formation and only a slight increase in resistance, while Str and dihydrostreptomycin (Dhs) showed almost no reduction and little resistance. Overall, the strong correlation between error cluster reduction and resistance, together with our biochemical, kinetic, and structural data, support the conclusion that EF-G resistance variants selectively silence corrupted ribosomes, thereby reducing the proteotoxic stress (Supplementary Fig. 2, model 5).

### *fusA* mutations help to maintain proteostasis and membrane integrity

Because the proposed resistance mechanism is most effective at low intracellular AGA concentrations, we investigated how proteostasis and membrane integrity are affected in *fusA* mutant strains at extracellular AGA concentrations that are lethal for wt bacteria. Light microscopy revealed significant protein aggregation at the poles of Apr-treated wt cells (Fig. 6a), which is consistent with previous microscopic studies of AGA-treated cells[11,52]. These aggregates form due to AGA-induced misreading, which leads to protein misfolding and the activation of the unfolded protein response[12,53]. The unfolded protein response then triggers the expression of the small chaperones IbpA and IbpB, which guide unfolded proteins into aggregates[52,54]. In contrast, *fusA* mutant cells appear unaffected by Apr treatment (Fig. 6a and Supplementary Fig. 9). Supporting our microscopic analysis and error burden evaluation (Figs. 4b, c and 5a), the quantification of these chaperones, along with their transcription factor RpoH, revealed that the unfolded protein response is strongly induced in Apr-treated wt cells but remains unaffected in resistance mutant strains (Fig. 6b), supporting the notion that *fusA* mutations confer resistance by maintaining proteostasis.

We then investigated whether *fusA* mutants affect AGA uptake. First, we assessed the membrane integrity of Apr-treated wt and P610L cells by staining them with propidium iodide, a membrane-impermeable dye. While wt cells took up the dye, indicating membrane damage, P610L cells remained unstained, suggesting that *fusA* mutations help preserve membrane integrity after AGA treatment (Fig. 6c). Second, to investigate the specific effect on AGA uptake, we treated wt and mutant strains with Apr and then used gentamicin-Texas Red (GTTR) as a reporter to visualize AGA uptake (Fig. 6d, e). Wt cells showed a strong increase in GTTR fluorescence after AGA treatment, indicating substantial antibiotic accumulation. In contrast, *fusA* mutant cells showed little or no increase in GTTR fluorescence during AGA treatment, suggesting low AGA uptake (Fig. 6e). These findings suggest that *fusA* mutations, by silencing AGA-bound ribosomes, contribute to resistance by limiting AGA uptake.

To test whether reduced antibiotic uptake explains the resistance phenotype of *fusA* mutants, we measured MIC values in the absence or presence of $AgNO_3$, a compound that permeabilizes bacterial membranes and accelerates AGA uptake[55] (Fig. 6f). While the *fusA* mutant strains appeared to maintain a low-level residual AGA resistance after $AgNO_3$ treatment, the differences to the wt strain were not statistically significant (one-way ANOVA with Šidák correction). The large (25-50-fold) effect of $AgNO_3$ strongly supports the notion that while selective silencing of corrupted ribosomes is the primary microscopic effect of

*fusA* mutations, the preservation of the membrane integrity is the dominant macroscopic outcome that ultimately enables the cell to survive at otherwise lethal AGA concentrations.

### Discussion

Our findings reveal a previously unrecognized resistance mechanism: EF-G resistance variants can efficiently promote translation in the absence of AGAs, but selectively silence AGA-bound ribosomes allowing the drug to dissociate before the ribosomes decode the next codon, resulting in fewer translation errors (Fig. 7). Repeated over multiple slow elongation cycles, this mechanism reduces the formation of error clusters, particularly of highly destabilizing long error clusters with multiple amino acid substitutions, which are especially toxic[56]. The resulting lower error burden helps *fusA* mutant strains to preserve the proteome and membrane integrity, attenuating the self-promoted AGA uptake and keeping intracellular AGA concentrations low. Lower AGA-induced misreading may also reduce the burden on cellular quality control systems (e.g., chaperones and proteases), enabling the cell to remove remaining faulty macromolecules. Overall, these adaptations enable the cell to avoid the disastrous AGA-induced downstream effects such as oxidative stress and to maintain normal growth despite exposure to otherwise lethal drug concentrations.

This mode of resistance clarifies the differential response to distinct AGAs. Resistance to Apr, KanA, KanB, Rib, and Sis correlates with the suppression of error cluster formation (Fig. 5f), and these AGAs lead to the selection of *fusA* mutations across organisms, despite their structural differences (Supplementary Fig. 1a,d and Supplementary Data 1). This highlights the critical link between the effects of *fusA* mutations on formation of error clusters and their associated resistance phenotype. In contrast, Nea, which binds to the same site but likely dissociates faster[57], does not induce error clusters[12] and elicits minimal *fusA*-mediated resistance (Fig. 1c). Consequently, Nea selects fidelity-enhancing mutations in ribosomal proteins uS5, uS12, and uS17 (Supplementary Fig. 1a). Similarly, Str-treated cells acquire mutations in uS12 instead of *fusA* mutations, consistent with its minor effect on translocation[58], lack of resistance (Fig. 1c), and lack of error cluster formation suppression in P610L (Fig. 5f). Likewise, Spc, which does not promote misreading, fails to select *fusA* mutations as well, which is in line with our proposed mechanism.

Although AGA resistance mutations are broadly distributed across EF-G (Supplementary Data 1), they do not occur randomly, and not all of them selectively silence AGA-bound ribosomes. Some less frequent resistance variants, such as G117C, R371L, and T674A, generally slow translocation (Fig. 2d), likely also increasing the chance of AGA dissociation and error clusters reduction. However, this general slowdown presumably comes at a higher fitness cost, likely limiting their prevalence. Similarly, *fusA* mutations that confer resistance to bacillaene[59] or fusidic acid[60] by preventing these drugs from binding to EF-G, also slow down translation and can confer cross-resistance to AGAs; however, these mutations are typically not selected under AGA exposure. In addition to reduced protein production, a general translation slowdown is more likely to cause ribosome stalling, triggering translation termination by ribosome rescue pathways[61], or promoting ribosome frameshifting[49,62]. This may explain why mutations of key catalytic residues, which could substantially slow down translation (e.g., H583 in *E. coli*[49]), are transiently sampled but not maintained[41].

In contrast, domain IV mutations (residues 585-610) avoid these fitness defects. Our structural data reveal how corrupted ribosomes are selectively stalled and explain why spontaneous frameshifting is prevented. Normally, wt EF-G rapidly binds to the ribosome and hydrolyzes GTP to GDP and Pi, driving the tRNA movement on the ribosome[63]. The subsequent Pi dissociation from EF-G triggers a conformational rearrangement of the ribosome complex, promoting mRNA-tRNA displacement and the movement of EF-G domain IV into

 

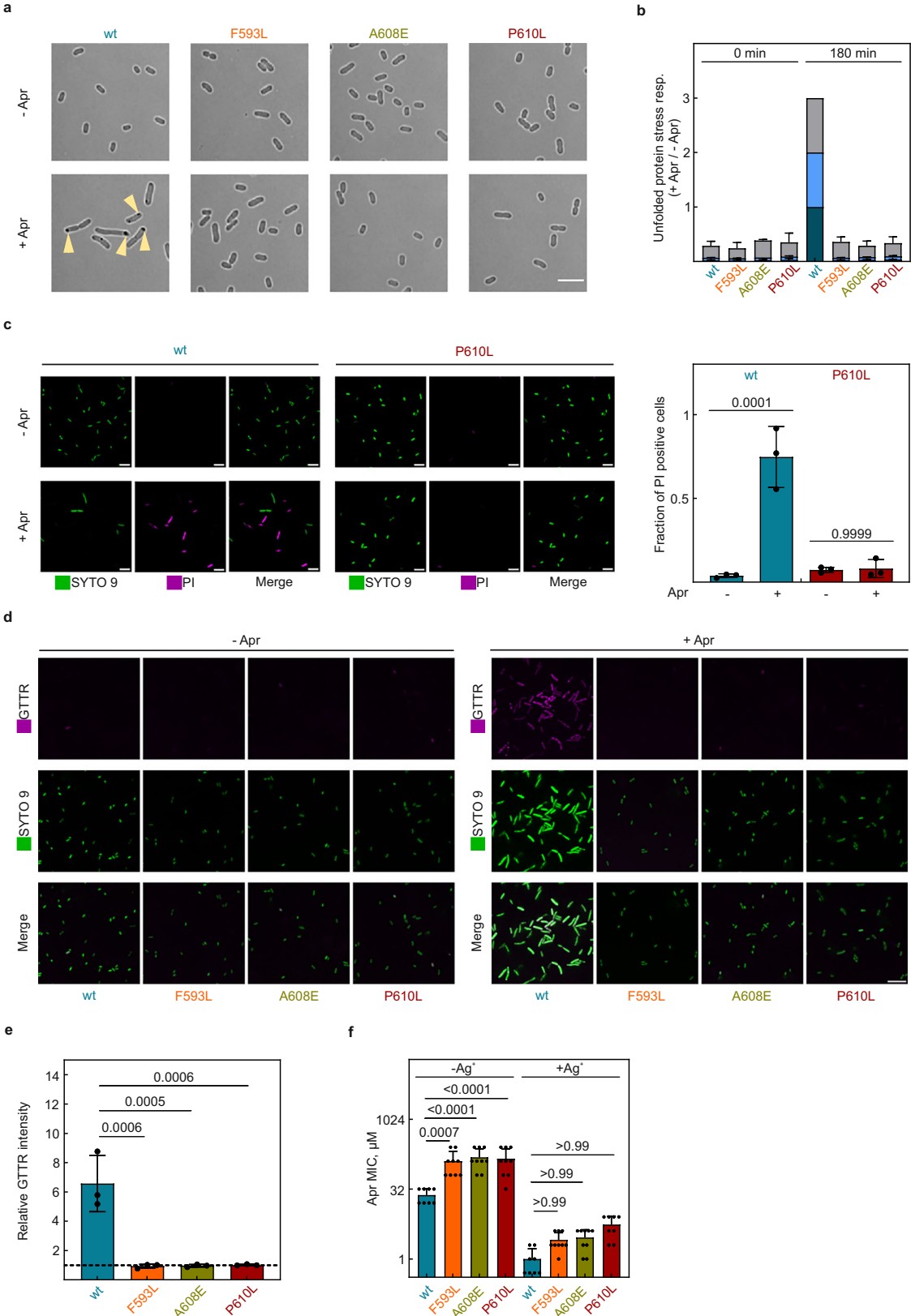

the A site of the ribosome. AGA binding alters the dynamics of the decoding center, stabilizing the mRNA-tRNA codon-anticodon complex in the A site and increasing the energy barrier for translocation[45,64]. While wt EF-G can overcome this barrier, it does so with a significant delay in translocation. For EF-G resistance variants, this added energy barrier poses a significant challenge, resulting in a translocation block at an early stage, before the mRNA and tRNA move on the small ribosomal subunit (Fig. 3). Halting translation at the early stage preserves the stabilizing interactions between the ribosome decoding center and the mRNA-tRNA codon-anticodon complex, preventing ribosome slippage. At the conditions where only a minor fraction of ribosomes is corrupted by AGAs, this mechanism likely

**Fig. 6 | *fusA* mutations preserve the membrane integrity and delay AGA uptake. a** Protein aggregation. Brightfield images of wt and *fusA* mutant cells treated with Apr (16 μM) for 180 min prior to imaging. Yellow arrows point to inclusion bodies formed in the wt cells. Scale bar, 5 μm. Representative images of 3 biological replicates. **b** Unfolded protein stress response. The expression of small chaperones IbpA/B involved in guiding misfolded proteins into aggregates and their corresponding transcription factor RpoH was quantified by targeted MS (label-free PRM using isotope-labeled reference peptides for identification). For each protein, abundance ratios (Apr-treated/untreated) were normalized to the wt 180 min. Bars represent mean values ± SD of 3 biological replicates (*n* = 3). **c** Membrane integrity. Cells were grown in the absence or presence of Apr (16 μM) for 180 min and the uptake of the membrane-permeable dye SYTO 9 and of membrane-impermeable propidium iodide (PI) was quantified by confocal microscopy. Representative images of 3 biological replicates (left panel; scale bar, 5 μm) and quantification (right panel). Bars represent the mean fraction ± SD of PI stained cells from 3 biological replicates (*n* = 3). Statistical significance was determined using a two-tailed Student's *t* test with Bonferroni correction; adjusted *P* values are indicated in

the figure. **d** AGA uptake. Representative Airyscan super-resolution images of wt and *fusA* mutant cells ± Apr (16 μM, for 90 min) and stained with GTTR (9 μM, 90 min staining). Shown images representative of 3 biological replicates. Scale bar, 8 μm. **e** Quantification of AGA uptake. Fold change in GTTR fluorescence intensity was measured in Apr-treated samples relative to the untreated control for wt and three *fusA* mutant strains. Bars represent mean GTTR fold change ± SD of 3 biological replicates (*n* = 3). Statistical significance was determined using a two-tailed Student's *t* test with Bonferroni correction for differences in mean GTTR fold change from wt only; adjusted *P* values are indicated in the figure. Dotted line indicates no change in fluorescence. **f** Impact of AgNO$_3$-mediated membrane permeabilization on Apr resistance. MIC values were measured in the absence and presence AgNO$_3$ (4 μM) by broth microdilution. Bars represent mean values ± SD of ≥8 biological replicates (exact *n* per bar are specified in the Source Data file). Statistical significance was determined by one-way ANOVA with Šidák correction; adjusted *P* values are indicated in the figure. Source data are provided as a Source Data file.

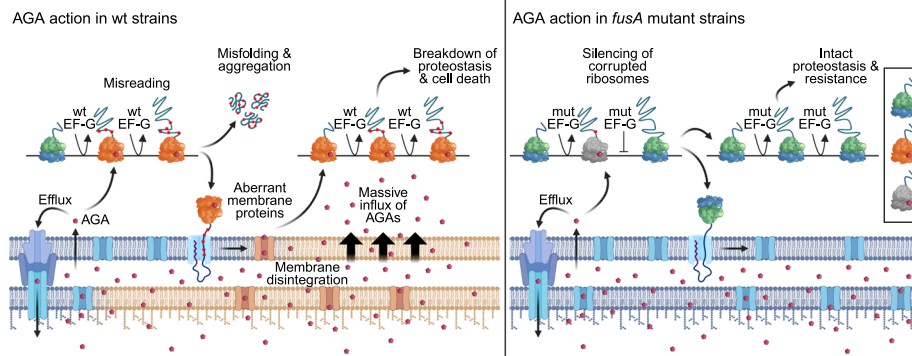

**Fig. 7 | Mechanism of AGA action in *fusA* mutant strains. Left panel: AGA action in wt strains.** At the early stages of AGA treatment, AGA enter the periplasm, but only small amounts seep into the cytoplasm where they bind to a few ribosomes which become corrupted. Corrupted ribosomes produce aberrant cytosolic and membrane proteins, leading to membrane disintegration and self-promoted massive influx of AGA. As more ribosomes become corrupted, high error load in newly synthesized proteins leads to a collapse of proteostasis, metabolic and ionic stress, and membrane voltage dysregulation, ultimately leading to cell death. **Right panel:**

**AGA action in *fusA* mutant strains**. As in the wt strain, initially only a few ribosomes become corrupted by AGA. These ribosomes are selectively silenced by the resistance EF-G variant until the drug dissociates, thereby lowering the error burden. This keeps the membrane intact, prevents self-promoted AGA uptake and maintains proteostasis. In addition, slower AGA uptake may allow the cell to clear AGAs from the cytosol, repair or remove damaged macromolecules, adapt to different lifestyles, or evolve additional resistance mutations resulting in AGA resistance. Created in BioRender (https://BioRender.com/shb0fhy).

minimizes the metabolic costs associated with globally slower translation and reduces undesirable side effects on ribosome function.

Despite the benefits of domain IV mutations and the high sequence conservation of *fusA*, resistance patterns vary among pathogens. One likely reason for this is that, apart from the individual AGA and the selection regime, resistance mutations will depend on the specific growth conditions. In *P. aeruginosa, fusA* mutations mainly arise in cystic fibrosis patients under long-term selection and counterselection, generating diverse mutation landscapes. Often, different populations of *fusA* mutations evolve even within the lungs of individual patients[58,65]. Species-specific traits also influence the selected mutations: while the mechanism involving silencing of AGA-bound ribosomes appears predominant in *E. coli*, a dedicated translation attenuation mechanism triggers the subsequent expression of MexXY efflux pumps in *P. aeruginosa*[66,67]. Although *E. coli* lacks this regulatory pathway, our proteome analysis suggests that slower translation caused by *fusA* mutations may in principle trigger attenuation pathways (e.g., in the Leu, Ile, and Trp operons) (Supplementary Fig. 4b, f). In such mechanisms, the mRNA regulatory elements, AGAs, and EF-G variants might act synergistically, resulting in distinct effects of *fusA* mutations on attenuation at specific mRNAs and on general misreading, thus affecting the selection of predominant resistance mutations. Therefore, we speculate that although the mechanics of EF-G action on the ribosomes resulting in selective silencing of corrupted ribosomes might be conserved, the downstream consequences might be species-

and sequence-dependent. Consequently, the general reduction of the error burden might be accompanied by dedicated resistance programs.

Classical AGA resistance enzymes preserve proteostasis by altering either the drug or the ribosome to prevent AGA binding[68]. We show that *fusA* mutations can achieve a similar outcome indirectly by preventing AGA uptake. However, these two antibiotic resistance mechanisms are conceptually distinct in the broader context of combating antibiotic resistance. Drug-modifying enzymes confer resistance by modifying specific functional groups of AGAs. Consequently, selecting AGA that lack these groups[69] or chemically derivatizing the drugs can restore their efficacy. In contrast, *fusA* mutations impact the mechanics of ribosomal translation itself, providing a more general resistance mechanism against a wide set of structurally diverse AGAs. Thus, evolution of *fusA* mutations provides a broader resistance mechanism against diverse AGAs. For example, strains harboring widespread resistance cassettes remain sensitive to Apr, a promising drug candidate that is less ototoxic and lacks functional groups common to other AGAs[69,70]. However, because Apr kinetically affects translation similarly to other AGAs, *fusA* mutations can silence Apr-bound ribosomes, thereby compromising its efficacy. Similarly, the activity of newly designed Tob derivatives, which have shown potency against AGA-resistant clinical isolates of *P. aeruginosa*, was compromised by *fusA* mutations[71]. However, *fusA* mutations do not confer uniform resistance to all AGAs. This suggests that a deeper

understanding of their resistance mechanism might inform the development of new AGAs in the future.

The selective silencing mechanism may extend to other ribosomal protein mutants. Potential candidates include error-restrictive uS12 variants, which not only reduce tRNA selection speed, improving translation fidelity, but also slow down translocation, resulting in reduced cell growth[72–74]. In addition, C-terminally truncated uL6 variants confer AGA resistance through an unknown mechanism in different bacteria[75–77]. In *S. aureus*, small-colony-forming variants that survive AGA treatment often have mutations in *fusA* or *rplF*[77], which encodes uL6, suggesting that their resistance mechanisms might be similar. The C-terminal domain of uL6 plays a dual role: it binds the incoming aminoacyl-tRNA early in tRNA decoding[78] and interacts with domain V of EF-G during translocation (Supplementary Fig. 6g). Disruption of these interactions may silence corrupted ribosomes similarly to *fusA* mutations. However, in a cellular context, this may not be favored, as *rplF* mutations or deletions are often associated with ribosome assembly defects and significant growth defects[76,77,79].

In summary, we show that *fusA* mutations silence AGA-bound ribosomes, preserving membrane integrity and attenuating AGA uptake. This strategy minimizes translation errors without modifying the ribosome itself—a key advantage, as altering the ribosome could affect all stages of translation, including ribosome assembly. While demonstrated in *E. coli*, the mechanism offers a framework for understanding evolutionary resistance patterns in the translation systems of various bacteria, including ESKAPE pathogens, suggesting broad applicability across diverse bacterial species. Beyond AGAs, other bactericidal antibiotics—such as beta-lactams, griselimycins, macrolones, fluoroquinolones, or rifampicin—corrupt their targets, rendering them into killing devices[80–82]. Thus, the novel concept of silencing corrupted targets may offer valuable insights into resistance mechanisms across multiple antibiotic classes.

## Methods

### Chemicals and isotope-labeled reference peptides
Unless otherwise specified, chemicals were purchased from Merck or Sigma. Chemicals used for LC-MS/MS were of HPLC/MS grade and obtained from Thermo Fisher. Samples were handled in low-retention reaction cups (Eppendorf). For the reliable identification of missense peptides, isotope-labeled reference peptides (spikeTidesL, JPT) were used where indicated. Contamination with unlabeled peptides in the reference standards was monitored and found to be negligible at the low concentrations of reference peptides used.

### Bacterial strains and cell growth
A list of all strains and constructs used in this study can be found in Supplementary Information. Sequences of the synthetic regions and of the oligonucleotides used for sequencing are summarized in the Source Data file. *E. coli* reference strain MG1655 was purchased from the German Collection of Microorganisms and Cell Cultures (Braunschweig, Germany). Construction of EF-G variants F593L, A608E, and P610L in the essential *fusA* gene of *E. coli* MG1655 was performed according to Kim et al., with minor changes[83]. Red/ET recombination was used to replace 339 bp of the wt *fusA* gene starting from amino acid F593 with a codon-optimized part of *fusA* in combination with a Flippase Recognition Target (FRT)-flanked selection marker (Zeocin™). The wt part of *fusA* and the artificial codon optimized part generate a *fusA* gene chimera, encoding for exactly the same amino acid sequence as the wt *fusA* gene. The three *fusA* variants were generated in the same manner but using codon-optimized DNA fragments encoding for the desired mutations at amino acid positions 593, 608, and 610 (Supplementary Fig. 3a). All clones were analyzed by sequencing of the modified region.

*E. coli* cells were grown in LB medium at 37 °C. In general, 500 ml cultures were grown at 200 rpm to $OD_{600}$ of 0.2-0.3 and treated with

AGAs for 60 min, 120 min, or 180 min as indicated. To determine doubling times, cells were inoculated to $OD_{600}$ of 0.01 and cell growth was recorded between 0.05 and 0.5 by measuring the $OD_{600}$ over time and fitting the data to the Malthusian growth equation.

MIC values were determined by broth microdilution. Briefly, single bacterial colonies were used to inoculate overnight cultures, which were then diluted to an initial optical density of 0.0005 ($OD_{600}$) in LB medium. Twofold serial dilutions of the tested antibiotics were prepared in 96-well plates, maintaining a final volume of 200 µl medium per well. Plates were incubated with shaking at 400 rpm in a Thermomixer (Eppendorf) at 37 °C for 22 h. Bacterial growth was monitored using a FERAstar FS plate reader (BMG Labtech) and MIC values were defined as the lowest antibiotic concentration that inhibited cell growth by ≥90% compared to untreated controls.

### Purification of inverted inner membrane vesicles
To detect amino acid substitutions in inner membrane proteins, it was necessary to first enrich the membrane protein fraction. This was achieved by generating inverted inner membrane vesicles. The process involves mechanically lysing *E. coli* cells, followed by density gradient centrifugation to separate inverted outer and inner membrane vesicles. In detail, bacteria were grown in the absence and presence of Apr (16 µM, 60 min) as described above and harvested by centrifugation at $5000 \times g$ for 10 min. Cell pellets were flash frozen in liquid nitrogen and stored at −80 °C. Cell pellets were thawed in buffer A (50 mM Tris-HCl pH 7.8, 300 mM NaCl, 10% glycerol (v/v), 5 mM 2-mercaptoethanol containing cOmplete protease inhibitor (1 tablet per 50 ml, Roche Diagnostics)). Cells were lysed using the EmulsiFlex C3 (Avestin). Cell debris was removed by centrifugation. Inverted inner membrane vesicles in the supernatant were collected by ultracentrifugation at 148,000 g (35,000 rpm in Ti50.2 rotor, Beckman Coulter) for 1 h. Membrane vesicles were resuspended in 20% (w/v) sucrose in 20 mM Tris-HCl pH 7.8 using a tissue grinder and inner and outer membrane vesicles were separated by ultracentrifugation through a sucrose step gradient (73% (w/v) and 53% (w/v) in 20 mM Tris-HCl pH 7.8) at 90,000 g (23,000 rpm in a SW32 rotor; Beckman Coulter) for 20 h. Inner membrane vesicle fractions were pooled and inner membranes were collected by ultracentrifugation at 148,000 g (35,000 rpm in Ti50.2 rotor, Beckman Coulter) for 1 h. Membrane pellets were washed with 100 mM $Na_2CO_3$ and 6 M urea in 50 mM Tris-HCl pH 7.8. Inner membrane vesicle pellets were resuspended in buffer A and stored at −80 °C. For the quantification of missense peptides, inner membrane proteins were separated on an anykD Criterion gradient SDS PAGE (BIO RAD). Bands of abundant inner membrane proteins were stained with Coomassie and in-gel proteolyzed with trypsin (Sequencing grade modified, Promega)[84].

### Cloning and protein purification of EF-G variants
*fusA* from *E. coli* MG1655 was cloned into pET-24a (Novagen) and mutations were introduced by QuikChange PCR. The EF-G (wt and variants) were overexpressed in *E. coli* BL21(DE3) and cells for His-tagged EF-G purification were lysed in buffer B (25 mM HEPES-HCl pH 7.5, 400 mM KCl, 10% glycerol (v/v), 5 mM 2-mercaptoethanol containing cOmplete protease inhibitor (1 tablet per 50 ml, Roche Diagnostics) and traces of DNase I (Sigma Aldrich)) using an EmulsiFlex C3 (Avestin). Cell debris was removed by centrifugation and EF-G was purified using the Protino Ni-IDA 2000 kit (Macherey-Nagel) according to the manufacturer's protocol. EF-G eluted with 200 mM imidazole was rebuffered into buffer C (25 mM HEPES-HCl pH 7.5, 70 mM $NH_4Cl$, 30 mM KCl, 7 mM $MgCl_2$) and stored at −80 °C.

### In-vitro translocation: Complex preparation and translocation experiments
Ribosomes from *E. coli* (MRE600), initiation factors, EF-Tu, f[³H]Met-tRNA^fMet, [¹⁴C]Phe-tRNA^Phe, fluorescein-labeled mRNA(MF+14Flu), and

pre-translocation complex (PRE) were prepared and purified as described[49]. The mRNA used in the translocation experiments was purchased from IBA. The sequence of the model mRNA(MF) was: 5′-GUUAACAGGUAUACAUACU<u>AUG</u>UUUGUUAUUAC-3′ (start codon is underlined). Translocation experiments were carried out in buffer D (50 mM Tris-HCl pH 7.5 at 37 °C, 70 mM $NH_4Cl$, 30 mM KCl, 7 mM $MgCl_2$). mRNA translocation was measured by monitoring the fluorescence change of MF+14Flu. The MF+14Flu-programmed PRE complexes (0.05 μM) were mixed with EF-G (wt or variants, 5 μM) and Apr or KanA (0 and 250 μM) in a stopped-flow apparatus (SX 20, Applied Photophysics) at 37 °C. Fluorescein was excited at 470 nm and emitted light was detected after passing a KV500 cut-off filter. The changes in fluorescence were recorded over 10-100 s and 3-10 replicates were measured for each condition. The rate of mRNA translocation was determined by exponential fitting using GraphPad Prism (version 10.2.3)[45].

## In-vitro translation

Initiation complexes with *slyD* mRNA were prepared in buffer E (50 mM HEPES-HCl pH 7.5, 70 mM $NH_4Cl$, 30 mM KCl, 7 mM $MgCl_2$, 2 mM DTT, and 2 mM GTP). Ribosomes (0.5 μM) were incubated with initiation factors (IF1, IF2, and IF3; 2.25 μM each), *slyD* mRNA (2 μM), and BodipyFL-[³H]Met-tRNA$^{fMet}$ (1 μM) for 45 min at 37 °C. Ternary complex EF-Tu−GTP−aa-tRNA was prepared in buffer E by incubating EF-Tu−GDP (120 μM) with phosphoenolpyruvate (3 mM), and pyruvate kinase (0.05 mg/mL) for 15 min at 37 °C, mixing with total aminoacyl-tRNA (200 μM), and further incubation for 1 min at 37 °C. In-vitro translation was performed in HiFi buffer (50 mM HEPES-HCl pH 7.5, 70 mM $NH_4Cl$, 30 mM KCl, 3.5 mM $MgCl_2$, 1 mM DTT, 0.5 mM spermidine, and 8 mM putrescine)[9]. Initiation complexes (80 nM) were mixed with EF-Tu−GTP−aa-tRNA (100 μM) and EF-G (wt or variants, 1 μM), and incubated at 37 °C without or with Apr. Translation products were separated by Tris-Tricine PAGE[85]. Fluorescent peptides were detected using Starion IR/FLA-9000 scanner (FUJIFILM) and quantified using Multi Gauge software (FUJIFILM). Uncropped scans of representative SDS-PAGES can be found in the Source Data file.

## Light and fluorescence microscopy methods

To visualize protein aggregates in vivo, 1 $OD_{600}$ of untreated and Apr-treated cells (16 μM, 180 min) were harvested by centrifugation and washed with 0.85% NaCl (w/v) in water. Cells were applied to 1.5% (w/v) low-melt agarose (Lonza) placed on 25 × 75 mm glass slide (Epredia) and covered with a cover glass (Menzel Gläser) for microscopy. Bright-field images were captured on a LSM 880 microscope (Zeiss) equipped with a plan apochromat 63x/1.4 NA oil objective. Minimum intensity was set to 112 for visualization and the scale bar was added using Fiji (NIH) software.

To assess the impact of *fusA* mutations on membrane permeability, untreated and Apr-treated cells (16 μM, 180 min) were harvested by centrifugation and washed twice with 0.85% NaCl (w/v). Cells (1 $OD_{600}$) were stained with SYTO 9 and propidium iodide stains from the LIVE/DEAD® BacLight™ Bacterial Viability Kit (Invitrogen) following the manufacturer's instructions. Briefly, equal volumes of SYTO 9 and propidium iodide stains were mixed and 1.5 μl of the mix was used to stain the cells for 15 min in the dark at room temperature. To remove excess stain, cells were washed with 0.85% NaCl (w/v). Cells were applied to low-melt agarose pads (as described above) for confocal microscopy. Confocal images were acquired on a Zeiss LSM 880 confocal microscope equipped with a 63x/1.40NA Plan Apochromat oil objective. Images were recorded using Zen Black 2.3 software and analyzed in Fiji (NIH) software. Cells were counted manually using the Cell Counter plugin of Fiji (NIH).

To monitor the AGA uptake, untreated and Apr-treated (16 μM, 90 min) cells were collected and incubated for 90 min at 37 °C with gentamicin Texas Red (GTTR, AAT Bio, 9 μM). Cells were washed twice with 0.5 ml of 500 μM unlabeled gentamicin (Sigma Aldrich), followed by 0.5 ml of LB medium and finally with 0.5 ml of 0.85% NaCl (w/v) to remove excess and non-internalized GTTR. Cells were stained with SYTO 9 stain from the LIVE/DEAD® BacLight™ Bacterial Viability Kit (Invitrogen) following manufacturer's instructions (15 min in the dark at room temperature). To remove excess stain, cells were washed with 0.85% NaCl (w/v). Cells were applied to low-melt agarose pads (as described above) for Airyscan super-resolution microscopy. Airyscan raw images were processed using Zen Black 2.3 software (Zeiss) using default settings. MicrobeJ plugin[86] (version 5.14p) of Fiji version 2.140 (NIH) was used to create closed cell contours around each cell based on the SYTO 9 signal. The mean GTTR fluorescence pixel intensity was measured within the given contour of each cell. The median intensity of all cells was calculated for each image. To compare fold changes, the ratio (Apr-treated/untreated control) of intensities was calculated using the averaged median fluorescence values from three images from three biological replicates. Statistical significance was determined using a two-tailed Student's *t* test with Bonferroni correction for differences from wt only; ***$P_{adj} < 0.001$. Maximum fluorescence values of representative images were readjusted for visualization, not for quantification; GTTR channel: 20,000, SYTO 9 channel: 24,000. All microscopy experiments were replicated thrice with three independent fields of view acquired per replicate.

## Cryo-EM analysis

Cryo-EM grids were prepared at 4 °C in a time-resolved manner with a total time of ~10 s to vitrification after mixing of EF-G P610L−GTP with PRE complex, using identical conditions as described for wt EF-G[51]. Specifically, 3 μL of 0.8 μM Apr-treated PRE complex were mixed with 3 μL of 2 μM EF-G P610L−GTP in buffer F (50 mM HEPES-HCl pH 7.5, 70 mM $NH_4Cl$, 30 mM KCl, 3.5 mM $MgCl_2$, 50 μM Apr, 0.6 mM spermine, 0.4 mM spermidine and 1 mM GTP), resulting in the final concentrations of 0.4 μM PRE and 1 μM EF-G P610L. Ribosome−EF-G P610L complexes were then immediately applied to glow-discharged EM grids (Quantifoil 2.7/1.3 μm), manually blotted for 8-10 s using What-man® Grade 1 Qualitative Filter Paper, and plunge-frozen in liquid ethane at 4 °C and 95% humidity using a custom-made device.

Cryo-EM data acquisition was performed on a Titan Krios electron microscope (Thermo Fisher Scientific) operating at 300 kV with an XFEG electron source, a spherical aberration corrector (CEOS Heidelberg), and a Falcon III direct electron detector (Thermo Fisher Scientific). Images were recorded using EPU v.2.3 in movie mode with an electron dose of 40 ± 5 e-/Å², a defocus range of −0.3 to −1.2 μm and a nominal magnification of 59,000×, yielding a final pixel size of 1.16 Å.

Data processing was performed using RELION v. 3.1 and 5.0[87] (Supplementary Fig. 10a), if not mentioned otherwise. Image stacks were motion-corrected with MotionCor2[88], and CTF parameters were estimated with CTFFIND v.4.1[89]. Particle selection was performed automatically using Gautomatch v. 0.56 (K. Zhang, MRC-LMB, Cambridge), followed by extraction and subsequent processing in RELION 3.1 (step 1 in Supplementary Fig. 10a). Selected particles were sorted for particle quality by 2D classification at a binned pixel size of 4.64 Å, resulting in an initial set of 1,753,901 ribosome particle images (step 2). 3D classification without alignment for particle quality and global ribosome conformation yielded two populations showing rotated and non-rotated ribosomal subunits, respectively (step 3). Further image processing was carried out separately for the two groups at the final pixel size of 1.16 Å. Per-particle motion correction was performed using the Bayesian polishing approach[90], followed by CTF refinement[91] for per-particle defocus and per-micrograph astigmatism. Subsequently, another round of Bayesian polishing and CTF refinement was performed. In this round, the CTF refinement was executed on a 3 × 3 grid to account also for off-axial aberrations, correcting first for magnification anisotropies, and then for per-particle defocus, per-micrograph astigmatism, beamtilt, trefoil and fourth-order aberrations.

The population of rotated ribosomes was sorted by global 3D classification (step 5 in Supplementary Fig. 10a), which allowed us to identify pre-translocation state ribosomes with both A- and P-site tRNAs bound, and to remove ribosomes with only P-site tRNA, isolated LSUs and low-quality particles. All subsequent sorting steps were performed using focused classification with signal subtraction (FCwSS). Sorting on the A-site tRNA resulted in two ribosome populations (step 6), hybrid state group 1 (H1 group) and hybrid state group 2 (H2 group), which differed in the orientation of the elbow region of the A-site tRNA on the LSU[51]. The H1 group was then sorted on presence of EF-G P610L and A-site tRNA (step 7), resulting in populations of ribosomes without (H1) and with EF-G P610L bound [H1-EF-G P610L], respectively. The particles containing EF-G P610L, were further sorted for EF-G domains 4 and 5 (step 8), yielding two sub-populations with different orientations of domain 4 ('D4-1' and D4-2' in Supplementary Fig. 10a). For the H2 group, FCwSS on the EF-G P610L core (step 6) identified a population of EF-G-free ribosomes (H2), as well as ribosomal complexes with EF-G P610L (H2-EF-G).

The group of non-rotated ribosomes was first sorted by global 3D classification for particle quality and presence of tRNAs, followed by FCwSS focusing on the tRNAs (step 4 in Supplementary Fig. 10a), resulting in a final population of non-rotated ribosomes with tRNAs bound in classic pre-translocation states, A/A and P/P, respectively. Further extensive sorting trials, both on rotated and non-rotated ribosomes, did not reveal any complexes with EF-G P610L in a state after Pi-release and tRNAs in chimeric state, as observed previously for complexes with wt EF-G[51].

All final particle populations were refined to high resolution following the gold-standard procedure[87]. The final cryo-EM maps were post-processed by global sharpening in PHENIX v. 1.20.1[92] and low-pass filtered to the respective final resolutions (Supplementary Fig. 10b, c and Supplementary Table 1). Cryo-EM maps of EF-G P610L-bound ribosome complexes, H1-EF-G P610L and H2-EF-G P610L, were resampled to a pixel size of 0.6525 Å for improved visualization and atomic model refinement.

### Atomic model refinement

Initial atomic models for H1-EF-G P610L and H2-EF-G P610L states were created by fitting the atomic coordinates of the corresponding complexes with wt EF-G (PDB entries: 7PJV and 7PJW, respectively)[51] as rigid bodies into the cryo-EM density maps using ChimeraX v. 1.8-1.10.1[93]. The P610L residue substitution was introduced and minor structural refinements were done in WinCoot v. 0.9.8.95[94]. Subsequent atomic model refinement was performed in PHENIX 1.20.1[92]. Secondary structure and metal coordination restraints for atomic model refinement were prepared with phenix.ready_set, and real-space refinement was performed using phenix.real_space_refine with automated weighting over 300 iterations and five macrocycles. Two such rounds of refinement were conducted for each state, with manual optimizations of the structural models in WinCoot between cycles (Supplementary Table 1).

### Proteolysis and chromatographic peptide separation

*E. coli* cells (1 OD$_{600}$) were lysed in 1x Laemmli sample buffer (Bio-Rad). For proteome analysis, proteins (corresponding to 0.1 OD$_{600}$) were desalted by a 10% Criterion TGX SDS PAGE (Biorad). For the analysis of individual proteins, proteins (corresponding to 0.1 OD$_{600}$ cells) were separated using Criterion TGX SDS PAGE (10% or anykD, Bio-Rad). Proteins were stained with Coomassie and in-gel proteolyzed with trypsin (Sequencing grade modified, Promega)[84]. Peptides were analyzed on a Vanquish Neo UHPLC system coupled to an Orbitrap Exploris 480 mass spectrometer (Thermo Fisher). Tryptic peptides were loaded on a PepMap Neo C18 trap column (5 μm particle size, 5 mm length, 300 μm inner diameter (Thermo Fisher). Bound peptides were eluted and separation was performed on a C18 capillary column (31 cm length, 75 μm inner diameter, Reprosil-Pur 120 Å, 3 μm (Dr.

Maisch GmbH)) at a flow rate of 300 nL/min with a acetonitrile gradient in 0.1% formic acid.

### Proteome analysis by data-independent acquisition (DIA)

Peptides eluting from the Vanquish Neo UHPLC system coupled to an Orbitrap Exploris 480 mass spectrometer (Thermo Fisher) were analyzed in positive mode over a 118 min runtime using a data-independent acquisition (DIA) method. Orbitrap resolution setting was set to 120,000 for MS and 30,000 for MS/MS (Full Width at Half Maximum (FWHM)). The MS scan range was 350-1650 *m/z*, with automatic gain control (AGC) targets were set to 300% for MS and 1000% for MS/MS. Maximum injection times were set to 20 ms for MS and 55 ms for MS/MS. Precursor fragmentation was performed using normalized higher-energy collision-induced dissociation (HCD) at 30%. MS/MS scans were acquired after each MS scan in 40 isolation windows (Source Data file). For each strain (MG1655, wt, F593L, A608E, and P610L) four biological replicates were acquired, each in two technical replicates.

DIA data were processed using Spectronaut (v. 17.1.22, Biognosys) with the spectral library-free directDIA workflow. Data were searched against the *E. coli* (K-12, MG1655) reference proteome from Uniprot (UP000000625, download on 03.04.2023, 4,401 protein entries). Trypsin/P was set as digestion enzyme, allowing up to two missed cleavages. For directDIA database search and data extraction, default settings provided by Biognosys were used.

Protein abundance estimates generated by Spectronaut (PG.ProteinQuantity) were analyzed using the Perseus software platform (v.1.6.15.0). Technical replicates were averaged, while biological replicates ($n = 4$) were kept separate for statistical analysis. Raw values were log$_2$-transformed, and a filtering threshold was set to retain proteins observed in at least 70% of all MS runs. Missing values were replaced with random numbers drawn from a normal distribution (default settings: width 0.3, and downshift 1.8). To identify significantly altered proteins in the *fusA* mutants relative to the wt, protein abundances were first compared pairwise between wt and each mutant respectively (Fig. 1d) using an unpaired two-sample Student's $t$ test (S$_0$ = 0.5; Benjamini-Hochberg FDR = 0.05) and the corresponding significance values are reported as $q$ values. Significance thresholds were set to q < 0.05−or q < 0.01 for higher stringency−and a minimum fold change of ± 2. In addition, to globally identify proteomic changes between all strains (Supplementary Fig. 4a–f), significantly altered proteins were identified by ANOVA statistics applying a permutation-based FDR (S$_0$ = 1; FDR = 0.05; 250 randomizations). The log$_2$-transformed intensities of significantly regulated proteins were $Z$-score normalized, and $Z$-scores of biological replicates were averaged. Co-regulated proteins were identified by hierarchical clustering based on their Euclidian distance and grouped into 5 clusters. Enrichment of biological processes within individual clusters was determined using Fisher's exact test.

### Targeted quantification of unfolded protein stress response

Due to the low abundance of small chaperones under unstressed conditions, we employed targeted mass spectrometry (PRM) to comparatively analyze their expression levels across biological states. Proteolysis and sample preparation was performed as described above. For each target protein (IbpA, IbpB, RpoH, EF-Tu, and uL10), a panel of at least two proteotypic, tryptic peptides was selected to serve as quantitative reporters. Peptides were separated using an Ultimate 3000 RSLC system (Thermo Fisher Scientific) coupled to an Orbitrap Fusion Lumos Tribrid mass spectrometer (Thermo Fisher Scientific). In detail, peptides were initially loaded on a C18 precolumn (2.5 cm, 150 μm ID, Reprosil-Pur 120 Å, 5 μm), eluted and then separated on a

C18 capillary column (31 cm, 75 μm ID, packed with Reprosil-Pur 120 Å, 1.9 μm) at a flow rate of 300 nl/min, with a 68 min linear gradient from 5 to 42% acetonitrile in 0.1% formic acid. PRM data acquisition was used for label-free quantification using isotope-labeled peptides for peptide identification. In PRM acquisitions, precursor ions were isolated using 1 $m/z$ isolation windows, AGC target set to standard, and maximum injection time set to 'auto' for fragmentation by HCD, with a normalized collision energy setting of 30%. MS/MS transients were acquired at resolution setting of 60,000 (FWHM) in the Orbitrap mass analyzer. Sets of interference-free fragment chromatograms were extracted at a resolution of 60,000 using the Skyline software and abundance differences were estimated based on the sum of the individual integrated fragment intensities. To account for potential sample loading variability, signal intensities were normalized to the levels of two constitutively expressed proteins which are unaffected by AGA treatment, EF-Tu and ribosomal protein uL10. Means of peptide intensity ratios (Apr treated / untreated) for each protein were compared between the different strains.

## Identification of missense peptides in survey runs

For the identification of missense peptides, data acquisition was performed in positive ion mode using a Top 30 data-dependent acquisition (DDA) method. Typically, MS spectra were acquired at a resolution setting of 120,000 FWHM over a range of 350-1600 $m/z$. The normalized AGC target was set to 300%, with a maximum injection time of 50 ms. Precursors with charge states 2-6 and intensities above 1e$^4$ were selected for fragmentation at an isolation width of 1.6 $m/z$. Fragmentation was carried out using HCD with a normalized collision energy of 30%. Precursors with undetermined charge states were excluded from fragmentation selection, and masses of fragmented precursors were dynamically excluded for 22 s with a mass tolerance of ± 10 ppm. MS/MS transitions were acquired at a resolution setting of 15,000 FWHM, using a normalized AGC target of 100% and a maximum injection time of 200 ms. Acquisition parameters optimized for individual experiments are reported in the Source Data file. To gain additional missense peptide identifications that are aligned to the other identification runs, MS gas-phase fractionation was performed (380-500; 500-580; 580-660; 660-740; 740-1200 $m/z$).

For the identification of missense peptides with single errors (illustrated in Supplementary Fig. 7a), raw data files were first processed using MaxQuant (version 2.0.1.0)[95]. Unless stated otherwise, standard software settings were applied. Protein identification was performed using the *E. coli* proteome as the reference database, along with a database of known lab contaminants. To identify partial peptide spectrum matches, we utilized the MaxQuant Dependent Peptide search algorithm[96], which defines a dependent peptide as an encoded peptide carrying an additional, localized delta mass. Dependent peptide identifications were subsequently filtered for missense peptides with single amino acid substitutions using a Python script modified from Mordret et al. (2019)[97]. Missense peptides whose delta masses could be attributed to known post-translational or chemical modifications were excluded based on the Unimod database[98]. Additionally, substitutions attributed to non-cognate codon-anticodon mispairing (involving two or three base mismatches) as well as atryptic peptides were excluded from analysis. Furthermore, a mass tolerance of 0.005 Da and a localization probability threshold of 0.95 were applied. Missense peptide candidates whose sequences are also found in unrelated proteins were excluded. Lastly, a FDR of 0.01 was applied using a target-decoy approach, and the remaining dependent peptide identifications were accepted as missense peptide identifications. For the exploratory comparison of misreading events in P610L and wt (Supplementary Fig. 8b), the MaxQuant search database was supplemented with missense peptides carrying X to K/R and K/R to X substitutions, which are not systematically identified using the Dependent Peptide approach. These peptides were identified using the PEAKS

software (version 10.5) with the SPIDER algorithm. An MS/MS peptide spectrum library was generated from validated missense peptide identifications and imported into the Skyline software for quantification by targeted MS or MS1 filtering.

## Selection of missense peptides as reporter peptides

To select suitable reporter missense peptides with single amino acid substitutions for quantification in targeted MS, MS/MS spectra of identified missense peptides were matched against PROSIT-predicted MS/MS spectra[99] and only matches with ratio dot products ≥0.7 were included in the analysis. Additionally, at least four fragment ions were required to confirm peptide identity, including fragments covering the position of the substituted amino acid. If reliable identifications of cognate peptides were not possible, all corresponding missense peptides were excluded. Similarly, identifications were excluded if they could not be assigned to a chromatographic peak (ion dot products ≤ 0.7). Peptides with missed cleavages were excluded from analysis. In order to set up a readout that reflects AGA-induced misreading, only missense peptides with low background levels in untreated cells and a > 3-fold induction upon AGA-treatment were considered for analysis. For a relative comparison of missense peptides with single errors across various biological states, peptides with a high signal intensity, moderate hydrophobicity, and good chromatographic behavior were selected.

For the analysis of missense peptides with error clusters, we chose candidates with two amino acid substitutions per peptide, which were initially identified using the PEAKS software using the SPIDER algorithm[12]. All missense peptides with error clusters were observed and validated previously. In general, AGA-induced error clusters were observed in various proteins and shared similar properties (such as their linear dependence on the first error)[12]. Thus, we selected missense peptides with error clusters from EF-Tu as the most abundant protein in the *E. coli* cytosol, because this allows their confident quantification in AGA-treated cells across multiple biological states even for *fusA* mutants with diminished levels of error clusters. Furthermore, we selected error clusters for which both errors were also observed individually in single substituted peptides and all three peptides were induced by AGA treatment > 5-fold. We further kept only candidates for which the net delta mass ≠ 0. From the list of all possible peptides with error clusters, we prioritized missense peptides with good ionizability and observability to ensure a suitable dynamic range of detection for analyzing error cluster reduction in *fusA* mutant strains. To enable reproducible quantification, we selected peptides with moderate hydrophobicity and good chemical stability. Furthermore, we chose missense peptides with amino acid substitutions that did not substantially destabilize EF-Tu[12] and varied in the distance between individual misreading events. All missense peptides that were selected for targeting and that were unambiguously identified by targeted MS using isotope-labeled peptides as reference were included in the analysis.

## Detection of missense peptides by Parallel Reaction Monitoring (PRM)

Samples were analyzed on a Vanquish Neo UHPLC system coupled to an Orbitrap Exploris 480 mass spectrometer (Thermo Fisher) as described above. Acquisition parameters were tuned for the individual experiments and performance of the mass spectrometer, and can be found in the Source Data file. In general, the highest populated charge states of correct and missense peptides were targeted. In general, sets of interference-free fragment chromatograms were extracted using the Skyline software. Because amino acid substitutions can alter the fragmentation pattern of peptides, the chosen set of extracted fragments can differ for correct peptides, single error peptides, and

peptides with error clusters. Because we interpret the difference between wt and mutants, these differences have no impact on our conclusions. Abundance differences were estimated based on the sum of the individual integrated fragment intensities.

## Quantification of missense peptides

For the quantification of missense peptides by spectral counting (Supplementary Fig. 7c), missense peptides were identified in DDA runs as described above and their corresponding MS/MS spectra were counted. For a more comprehensive analysis, all types of amino acid substitutions were included if their respective counts were induced >3-fold by AGA-treatment.

For the intensity-based, label-free quantification of missense peptides, we applied different acquisition and analysis strategies: First, for the exploratory profiling of misreading events at high error load (Supplementary Fig. 8b), we identified missense peptides as described above and quantified them using MS1 filtering in Skyline. Identifications were imported into Skyline, and precursor ion signals were extracted at a resolution of 60,000. Missense peptides were identified as the peptides with the highest scoring identification across all aligned chromatographic runs. Missense peptide abundances were estimated based on integrated MS1 signal relative to their correct parental peptide. Second, for the comparative analysis of missense peptides with single errors in EF-Tu and membrane proteins (Fig. 4 and Supplementary Fig. 7), we targeted missense peptides (see above) by PRM. Correct and missense peptides were identified by the high sequence coverage of their co-eluting ion fragments and their high dot product scores, reflecting the similarity between their fragmentation spectra and those derived from database search results (see above). Third, in all experiments involving the quantification of error clusters, we spiked in isotope-labeled reference peptides (Fig. 5a–f and Supplementary Fig. 8a) (JPT-L grade, JPT Peptide technologies, estimated <5 fmol on column). The reference peptides were not used for quantification but helped to select interference-free pseudo-transitions and to identify missense peptides by their close to one ratio dot product.

In general, peptide abundances were quantified by integration over consistent elution windows in all quantification runs. Integrated peak areas were exported from Skyline and error frequencies ($E_f$) were calculated as the intensity ratio of the missense peptide over the cognate peptide. $E_f^{next}$ values were calculated in two different ways. $E_f^{next}$ values derived from multiple biological states were estimated as the slope of the regression line when plotting the frequency of error clusters against the frequency of the initial misreading event (Fig. 5c). Alternatively, $E_f^{next}$ was estimated from point measurements at a fixed time and AGA concentration (Fig. 5a, d–f). Here, $E_f^{next}$ was estimated as the ratio of the abundances of the error cluster and the first misreading event.

In cases where fixed integration windows led to infinitely low error frequencies due to the absence of detectable peaks or noise, small constant values were imputed. Especially for long error clusters, detection in the *fusA* mutant strains was sometimes not possible due to their low abundance. Although in some cases this prevents the exact quantification of $E_f^{next}$ reduction, it clearly shows how strongly error clusters are reduced. Amino acid substitutions can alter the physico-chemical properties and ionization propensities of peptides significantly, thereby potentially affecting observed error frequencies. However, because we use error frequencies only for relative comparisons between wt and mutant strains, such effects have no impact on our conclusions. In a few cases—such as when Arg or Lys was substituted or introduced, altering the tryptic cleavage pattern—the corresponding parental peptide could not be detected. This was likely due to poor fragmentation or exclusion during the database search, for instance because of its small size. In these instances, the median intensity of correctly identified peptides was used as a reference. In general, similar amounts of target protein-derived peptides were analyzed and controls of non-treated cells were performed to exclude

false-positive identifications. Chromatographic carryover of peptides with misreading events between runs was evaluated and found to be below the detection limit of the mass spectrometer.

## Statistics and reproducibility

All central conclusions (e.g., EF-G variants silence AGA-bound ribosomes and *fusA* mutations shield bacteria against AGAs) are supported by orthogonal methods. All experiments were reproduced as stated in the figure legends. Sample sizes were selected based on standard practice in microbiology and proteomics to ensure reproducibility and statistical robustness. No statistical methods were used to pre-determine sample size. For MS, technical replicates are derived from repeats in the analysis of the same sample. Biological replicates derived from analysis of separate, biologically distinct samples produced independently of each other starting from culture inoculation from different clones, and followed by individual cell growth, drug treatment, sample processing, and data acquisition. When representative results are shown, the experiment has been repeated three times with similar results.

Quantitative proteomics data were analysed using MaxQuant (v2.1.4.0) with a FDR of 1% at the peptide and protein levels. Statistical testing for differential protein abundance was performed using two-sided Student's $t$ tests implemented in Perseus (v1.6.15.0), with significance thresholds and multiple testing corrections indicated in the respective figure legends. Missing values were imputed from a normal distribution as described above. For imaging experiments, image fields were chosen at random and differences in fluorescence intensity were evaluated using two-tailed Student's $t$ test. All attempts at replication were successful and no data were excluded from analysis.

Experiments were not randomized and investigators were not blinded, owing to the mechanistic nature of the study. Moreover, randomization and blinding were not applied because sample identity could be inferred from data structure, and key results (such as the reduction or length distribution of error clusters) can be independently derived from each individual dataset. All attempts at replication were successful and no data were excluded from analysis.

## Reporting summary

Further information on research design is available in the Nature Portfolio Reporting Summary linked to this article.

# Data availability

All data supporting the findings of this study are available within the article, its Supplementary Information, or in the indicated public repositories. The targeted MS data generated in this study have been deposited to the ProteomeXchange consortium via the Panorama partner repository under the accession code PXD063129. All proteomics data generated in this study have been deposited to the ProteomeXchange consortium via the PRIDE partner repository under the accession codes PXD061583 and PXD067947). Raw microscopy images have been deposited to Zenodo under the (https://doi.org/10.5281/zenodo.17087887. The cryo-EM maps/associated coordinates of atomic models generated in this study have been deposited in the Electron Microscopy Data Bank/Protein Data Bank under the following accession codes: EMDB: 55123; EMDB: 55124; EMDB 55125; PDB: 9RTU; EMDB: 54253 and PDB: 9RTV; EMDB: 54254. Previously published atomic coordinates from the Protein Data Bank (PDB accession codes 7PJS, 8CGJ, 8CEP, 8CA7, 8CF1, 7PJV and 7PJW) were used for illustrations and initial atomic model fitting. Source data are provided with this paper.

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

## Acknowledgements

This research was supported by the German Science Foundation (Deutsche Forschungsgemeinschaft, DFG) through SFB1565 (project-ID 469281184 to M.V.R. and I.W.) and Leibniz Prize (to M.V.R.), and by the Max Planck Society. We express our sincere gratitude to Holger Stark (MPI-NAT) for providing access to the infrastructure of his department at the MPI-NAT, Peter Lenart (Facility for Light microscopy) for helpful discussions on light miscroscopy experiments, and Michele Felletti (MPI-NAT, Department Rodnina) for critically reading the manuscript. We acknowledge Mario Lüttich and Tobias Koske (MPI-NAT, Department Stark) for support in high-performance computation and Ralf Pflanz (MPI-NAT, Bioanalytical Mass Spectrometry Group) for his help in mass spectrometry maintenance and acquisition. We thank Olaf Geintzer, Vanessa Herold, Franziska Hummel, Jasmin Jakobi, Sandra Kappler, Anna Pfeifer, Monika Raabe, and Michael Zimmermann for expert technical assistance.

## Author contributions

Conceptualization, M.V.R. and I.W.; chromosomal mutants, M.S.; MIC measurements, C.K.; in-vitro kinetics, B.Z.P. and E.S.; Cryo-EM, V.P., A.S., and N.F. using ribosome complexes from B.Z.P. and F.P.; mass spectrometry, N.G.D., N.S.F. and I.W.; light microscopy, N.G.D.; supervision, F.P., A.Z.P., H.U., N.F., M.V.R., and I.W.; N.G.D., N.S.F., M.V.R., and I.W. wrote the manuscript with input from all authors. All authors contributed to the interpretation of the data and approved the final manuscript.

## Funding

## Competing interests

The authors declare no competing interests.
