## [Transparent Peer Review file · Nature Communications]

Selective silencing of antibiotic-tethered ribosomes as a resistance mechanism against aminoglycosides

Corresponding Author: Dr Ingo Wohlgemuth

Version 0:

Reviewer comments:

Reviewer #1

(Remarks to the Author)

In this elegant study, Ghosh Dastidar et al. investigate how mutations in *fusA*, the gene coding for EF-G, confer resistance to aminoglycoside antibiotics. Using a combination of quantitative mass spectrometry, kinetic analyses, cryo-EM and live-cell imaging, the authors propose a mechanism by which *fusA* mutations selectively slow down elongation by drug-bound ribosomes. Although bacterial growth is only mildly affected by the mutations, the drug has more time to dissociate from the ribosome, resulting in fewer aminoglycoside-induced miscoding errors and decreased production of aberrant proteins compared to wild-type cells. This, in turn, preserves membrane integrity and prevents the self-promoted aminoglycoside uptake normally observed for this family of antibiotics, leading to resistance.

The identification of this novel resistance mechanism, termed “selective silencing of corrupted targets” is an important step forward in our understanding of antibiotic action and resistance. The study is well-designed, the conclusions are fully supported by the available data, and the results will be of interest to both the antibiotic resistance and translation fields.

Consequently, I only have the following minor comments for the authors to address:

- Lines 116-118. According to Fig. 1C, the extents of resistance to Nea and Neo appear to be very similar, even though error bars for Nea are larger. However the authors claim that the *fusA* mutant gives resistance to Neo (which induces error clusters), but not to Nea (which does not induce error clusters). This point should be clarified.
- Fig. 1b. The difference between the MG1655 and wt strains is explained in Supp. Fig. 3 and in the Methods section, but it would also be helpful to include this explanation on p. 4.
- Lines 244-251. It would be helpful to mention here that the structure obtained in this study was compared to an earlier structure that also contains Apr (Ref. 46). At present, this point is unclear and gives the impression that the Apr and EF-G P610L-containing structure is being compared to an EF-G wt structure lacking the drug.
- Fig. 3. Densities for the drug and for Domain IV of EF-G should be shown as supplementary figures. What changes (if any) does the mutation induce in EF-G?
- Line 333. Delete “the of”.
- Lines 366-381. How were specific mutation pairs chosen? Is the focus only on the most abundant peptides? In any case, the methodology should be briefly explained in the main text.
- Lines 345-346. Please specify the range of concentrations that “high Apr concentration” refers to.
- Line 448. The text should read “in the absence or presence of”.

Reviewer #2

(Remarks to the Author)

The work of Dastidar, Freyer, et al. presents a model to explain a resistance phenotype for aminoglycoside antibiotics (AGAs) whose mechanism had remained mysterious. Interpretation of the current data is based on previous work by the same group (Wohlgemuth et al, 2021) where it was elucidated that miscoding caused by AGAs results in translating proteins with clusters of erroneous amino acids, leading to proteotoxic stress and cell death. Here, based on compelling experimental evidence, authors propose how a bad trait, mutations in the EF-G encoding gene *fusA*, ends up being advantageous for AGA poisoned cells: the more sluggish AGA-bound ribosomes where translocation is driven by mutant EF-G, produce proteins with less error clusters and therefore, the proteome remains reasonably healthy and the membranes robust. The paper is clearly written, well-organized and the work is conceptually significant, using basic principles of translation to explain resistance to clinically-important antibiotics.

The following points of critique are mainly aimed to address specific points which are not clear in the manuscript:

- Fig. 1a: it won't be clear for many non-expert readers which portion of the ribosome is shown. Adding a panel with an overview of the whole AGA-bound ribosome interacting with EF-G and better color coding would help.
- Supplementary Fig. 2: the cartoon for the Gain of Function Model 4 is confusing as it shows that the translation aided by EF-G variants would result in proteins with more error clusters. How would this lead to resistance?
- Text is missing an explanation of the particles distribution shown in the bar graph of Fig. 3e and Supplementary Fig. 6 to clarify how the conclusions of which step is inhibited have been actually drawn.
- Related to the point above, legend of Fig. 3d: "in EF-G P610L the key reaction of Pi release is slowed down, inhibiting the conformational rearrangement required to promote the tRNA movement into the CHI state" – where does this come from?— from particle distribution shown in 3e?
- Fig. 3e: requires a comparison of the distribution of the same states but in the absence of Apr.
- It is not clear why mutant EF-G inhibits formation of the CHI state when AG is present
- The sentence in lane 285, "This contradicts the 'gain of function model'..." needs a better explanation.
- Ln 327-328- experiments assess relative error cluster formation, not rate of translation speed.
- p.11- Are the off rates of AGs bound to the translating ribosome known? If yes, is the model compatible with those rates?
- Would expression of wt EF-G overcome AGA resistance of the strains with mutations in chromosomal *fusA*?

Minor points:

- Revise the sentence on lanes 71-73: "...mutations...can confer high-level resistance, their impact on... resistance is limited..."
- ref. to Supplementary Fig. 2 in lanes 94-96 is misplaced.
- Lane 333, delete "of the".
- Lane 371, "than" not "that".
- Ln 436, correct "First, we first..."

Reviewer #3

(Remarks to the Author)

The manuscript by Dastidar et al. reports the results of an original integrative study poised at unraveling a new antibiotic resistance mechanism, which neutralizes the action of several aminoglycosides (AGAs), through certain mutations in the EF-G gene, *fusA*. The authors show that EF-G resistance variants slow ribosome movement along mRNA when aminoglycosides are bound in a selective fashion. The delay resulting from the slowing down increases the chance that the drug dissociates before misreading occurs.

The authors found that *fusA* mutations confer resistance early in treatment by preventing self-promoted aminoglycoside uptake, which occurs consequently to the accumulation of misreading errors on the translated membrane proteins/transporters.

Indeed, the accumulation of faulty membrane proteins and misreading-induced metabolic by-products makes the inner membrane permeable to AGAs. As more and more AGAs enter the cell, more ribosomes bind AGAs and become corrupted. Over time, this self-promoted uptake leads to a massive influx of AGAs, resulting in a burst of translation errors leading ultimately to the damage of macromolecular structures, the disruption of the general metabolism and the membrane voltage dysregulation, all leading to cell death.

AGA resistance has been greatly characterized over the past decades and most AGA resistance mutations occur outside of the ribosome itself. Unexpectedly, a major hotspot of AGA resistance mutations is the *fusA* gene. Interestingly, *fusA* mutations confer resistance in various bacteria as well as in parasites such as *Leishmania*.

Despite their clinical relevance, the mechanism by which *fusA* mutations confer resistance remained thus far unclear, as EF-G neither blocks AGA binding nor contacts the decoding center.

The authors successfully demonstrate the existence of this previously unrecognized resistance mechanism that selectively silences corrupted targets, by applying a combination of quantitative mass spectrometry (MS), live-cell imaging, kinetic analysis, and cryo-EM, thus revealing a novel AGAs resistance strategy with potential therapeutic implications.

The authors found that *fusA* mutations specifically counteract the effects of certain AGAs that both disrupt translocation and induce misreading with error cluster formation.

The authors focused on three mutations in domain IV of EF-G (F593L, A608E and P610L). First, Dastidar et al. show that the proteomic changes induced by these mutations were minimal. They have analyzed the in-vitro translocation activity of the purified EF-G variants in the absence of AGAs and show that translocation rates substantially decreased in the presence of AGAs, supporting the idea that these mutants are selectively hindered in promoting translocation on AGA-bound ribosomes

The authors went on and probed the structure of an EF-G variant (P610L) in complex with the ribosome and Apramycin (Apr), using single-particle cryo-EM. The resulting structure superimposes nearly perfectly with EF-G wt and shows no significant structural changes in the decoding center or the drug-binding site, supporting the notion that EF-G variants do not displace the drug from the ribosome. They also show that the early stages of translocation up to GTP hydrolysis are unaffected by Apr but the later steps appear to be inhibited, particularly Pi release and the tRNA movement into chimeric states.

In summary, the authors show that *fusA* mutations silence AGA-bound ribosomes, preserving membrane integrity, and attenuating AGA uptake. This strategy minimizes translation errors without modifying the ribosome itself.

The manuscript is well written and the figures are clear. The references are sufficient, as far as the reviewer could tell and the performed experiments are state of the art. Furthermore, the depth of the performed analysis is simply overwhelming.

The reviewer has no major points to raise and only has very few minor comments. The manuscript can be in principle accepted in its current state.

Minor comments

Line 168 "these adaptations are likely due to changes in translation rates"

AdaptATions, not adaptations

Why was SlyD used as an in-vitro translation model? Why not other mRNAs?

Figure 3 a-c:

Could the lack of significant changes in Apr binding on EF-G associated ribosomes, between the wt and the EF-G variant/mutant, rather be a consequence of the relatively modest resolution of the cryo-EM? Indeed, at 3Å many details can't be interpreted, such as solvent molecules and other subtle conformational changes with certainty...

Also, why is the resolution relatively modest at a time where structures of bacterial ribosomes are frequently below 2Å?

Why was Apramycin used or considered as the archetype AGA?

Reviewer #1 (Remarks to the Author):

In this elegant study, Ghosh Dastidar et al. investigate how mutations in *fusA*, the gene coding for EF-G, confer resistance to aminoglycoside antibiotics. Using a combination of quantitative mass spectrometry, kinetic analyses, cryo-EM and live-cell imaging, the authors propose a mechanism by which *fusA* mutations selectively slow down elongation by drug-bound ribosomes. Although bacterial growth is only mildly affected by the mutations, the drug has more time to dissociate from the ribosome, resulting in fewer aminoglycoside-induced miscoding errors and decreased production of aberrant proteins compared to wild-type cells. This, in turn, preserves membrane integrity and prevents the self-promoted aminoglycoside uptake normally observed for this family of antibiotics, leading to resistance.

The identification of this novel resistance mechanism, termed “selective silencing of corrupted targets” is an important step forward in our understanding of antibiotic action and resistance. The study is well-designed, the conclusions are fully supported by the available data, and the results will be of interest to both the antibiotic resistance and translation fields.

Reply: We thank the reviewer for the positive feedback on our manuscript and helpful comments.

Consequently, I only have the following minor comments for the authors to address:

- Lines 116-118. According to Fig. 1C, the extents of resistance to Nea and Neo appear to be very similar, even though error bars for Nea are larger. However the authors claim that the *fusA* mutant gives resistance to Neo (which induces error clusters), but not to Nea (which does not induce error clusters). This point should be clarified.

Reply: As stated in the main text (lines 117-119 of the revised manuscript), the *fusA* mutations do not confer significant resistance to Nea. While the effect size appears similar to Neo, only the effect of Neo is statistically significant. For Nea, even after increasing the number of biological replicates to $n=15$, the effect remained statistically non-significant.

- Fig. 1b. The difference between the MG1655 and wt strains is explained in Supp. Fig. 3 and in the Methods section, but it would also be helpful to include this explanation on p. 4.

Reply: Done. We included additional information on the wt strain into the main text (l. 101-105) and refer to the Method sections for the details on cloning strategy (the addition is marked red):

“To explore how *fusA* mutations confer resistance to AGAs, we constructed *E. coli* MG1655 strains harboring three frequently reported laboratory-evolved EF-G variants (F593L, A608E, and P610L) (Fig. 1a), which belong to a prominent cluster of resistance mutations in EF-G domain IV (Supplementary Data 1), **along with an isogenic wild-type control generated by the same cloning strategy but retaining the wt sequence** (see Methods, Supplementary Fig. 3a-d)”.

- Lines 244-251. It would be helpful to mention here that the structure obtained in this study was compared to an earlier structure that also contains Apr (Ref. 46). At present, this point is unclear

and gives the impression that the Apr and EF-G P610L-containing structure is being compared to an EF-G wt structure lacking the drug.

Reply: Done (l. 204-206). “Superimposing the Apr-bound ribosome-EF-G P610L complex with the previously published Apr-bound ribosome EF-G wt complex ⁵¹ shows no significant structural changes in the decoding center or drug-binding site (Fig. 3c).”

- Fig. 3. Densities for the drug and for Domain IV of EF-G should be shown as supplementary figures. What changes (if any) does the mutation induce in EF-G?

Reply: Done. In response to the reviewer’s and editor’s comments, Supplementary Fig. 6 has been split into two, with biological aspects shown in Supplementary Fig. 6, while technical aspects are presented in Supplementary Fig. 10). Densities for the drug (Supplementary Fig. 6b) and for domain IV of EF-G (Fig S6d) have been added. The overall cryo-EM densities for EF-G in H1 and H2 states show no significant differences between mutant (present study) and wt EF-G (Petrychenko et al. Nat Commun 2021). Domain IV is dynamic in all structures, as described in detail in the published paper. As the P610L mutation is located in the hinge region between domains IV and III/V of EF-G, we performed extensive sorting of this area, but could not detect clear differences in domain IV dynamics.

- Line 333. Delete “the of”.

Reply: Done (l. 254), thank you.

- Lines 366-381. How were specific mutation pairs chosen? Is the focus only on the most abundant peptides? In any case, the methodology should be briefly explained in the main text.

Reply: We agree that the details of error cluster analysis may not be fully clear from the main text (l. 242-270 of the revised manuscript). However, to maintain readability for a general audience, we prefer not to include extensive methods details in the Results section and instead refer readers to the Methods. In the revised Methods, we have rephrased the selection procedure to make it more specific and transparent, ensuring that the key aspects are now clearly traceable (l. 736-741):

“From the list of all possible peptides with error clusters, we prioritized missense peptides with good ionizability and observability to ensure a suitable dynamic range of detection for analyzing error cluster reduction in *fusA* mutant strains. To enable reproducible quantification, we selected peptides with moderate hydrophobicity and good chemical stability. Furthermore, we chose missense peptides with amino acid substitutions that did not substantially destabilize EF-Tu and varied in the distance between individual misreading events.”

- Lines 345-346. Please specify the range of concentrations that “high Apr concentration” refers to.

Reply: The term “high” was meant to refer here to all concentrations above which the error burden decreases again because the cells die faster than they can synthesize large quantities of defective proteins. In order to avoid the evaluative term “high,” we have now specified the exact

concentration range (l. 1197-1198): “The apparently lower error frequency in wt cells at high Apr concentrations $\geq 32\mu\text{M}$ is due to massive cell death and the rapid decline in actively growing cells.”

- Line 448. The text should read “in the absence or presence of”.

Reply: Done, thank you.

Reviewer #2 (Remarks to the Author):

The work of Dastidar, Freyer, et al. presents a model to explain a resistance phenotype for aminoglycoside antibiotics (AGAs) whose mechanism had remained mysterious. Interpretation of the current data is based on previous work by the same group (Wohlgemuth et al, 2021) where it was elucidated that miscoding caused by AGAs results in translating proteins with clusters of erroneous amino acids, leading to proteotoxic stress and cell death. Here, based on compelling experimental evidence, authors propose how a bad trait, mutations in the EF-G encoding gene *fusA*, ends up being advantageous for AGA poisoned cells: the more sluggish AGA-bound ribosomes where translocation is driven by mutant EF-G, produce proteins with less error clusters and therefore, the proteome remains reasonably healthy and the membranes robust. The paper is clearly written, well-organized and the work is conceptually significant, using basic principles of translation to explain resistance to clinically-important antibiotics.

Reply: We thank the reviewer for positive comments and appreciating the significance of the work.

The following points of critique are mainly aimed to address specific points which are not clear in the manuscript:

- Fig. 1a: it won't be clear for many non-expert readers which portion of the ribosome is shown. Adding a panel with an overview of the whole AGA-bound ribosome interacting with EF-G and better color coding would help.

Reply: We thank the reviewer for the suggestion and added the view of the complete ribosome with two tRNAs and EF-G in Figure 1a. The color scheme for tRNAs and EF-G has been harmonized with the rest of the manuscript.

- Supplementary Fig. 2: the cartoon for the Gain of Function Model 4 is confusing as it shows that the translation aided by EF-G variants would result in proteins with more error clusters. How would this lead to resistance?

Reply: The gain of function mechanism was proposed before error clusters were identified. We agree that Model 4, as originally illustrated, appears unlikely and would in fact lead to accumulation of error clusters. However, for the sake of completeness in our literature overview, we would like to keep the model. To present it in a more neutral way, we have revised the schematic and removed the strong error cluster formation.

- Text is missing an explanation of the particles distribution shown in the bar graph of Fig. 3e and Supplementary Fig. 6 to clarify how the conclusions of which step is inhibited have been actually drawn.

Reply: We now added an explanation in the text (now p. 6-7): “With EF-G P610L, we could not detect any particles in the chimeric state, despite extensive sorting of the cryo-EM data (Methods). Furthermore, we could also not find any post-Pi-release state that would correspond to a tRNA hybrid state with EF-G-GDP bound, as we found previously for wt EF-G⁵¹. Accordingly, with EF-G P610L the early stages of translocation up to GTP hydrolysis are unaffected by Apr but the later steps appear to be inhibited, particularly Pi release and the tRNA movement into chimeric states.”

- Related to the point above, legend of Fig. 3d: “in EF-G P610L the key reaction of Pi release is slowed down, inhibiting the conformational rearrangement required to promote the tRNA movement into the CHI state” – where does this come from?—from particle distribution shown in 3e?

Reply: Yes, please refer to previous point.

- Fig. 3e: requires a comparison of the distribution of the same states but in the absence of Apr.

Reply: In the present manuscript, we focus specifically on the impact of EF-G mutations on the particle distribution in the presence of Apr, and did not acquire cryo-EM data in the absence of Apr. Such experiments would be beyond the scope of the present manuscript, because in the absence of AGA, these EF-G mutants promote complete translocation at a rate comparable to that of wt EF-G (i.e., in milliseconds-seconds range), which makes cryo-EM experiments extremely tedious.

- It is not clear why mutant EF-G inhibits formation of the CHI state when AG is present

Reply: That is correct – our cryo-EM data clearly show that mutant EF-G blocks CHI state formation in the presence of AGA, but we do not yet understand the underlying mechanism. AGA binding to the ribosome greatly increases the energy barrier of translocation, possibly by reducing the dynamics of the decoding center. A further impairment of EF-G dynamics caused by mutation could uncouple ribosome and EF-G movements. At present, however, these remain speculative considerations that lead beyond the resistance focus of this paper, and we would prefer not to expand on them here.

- The sentence in line 285, “This contradicts the ‘gain of function model’ ...” needs a better explanation.

Reply: We thank the reviewer for this suggestion. We now more explicitly write (l. 218-220): “The fact that EF-G P610L and Apr can bind simultaneously, and that the mutation impairs late translocation steps is inconsistent with a ‘gain-of-function’ model (Supplementary Fig. 2, model

4). Together with the kinetic analysis (Fig. 2), these results demonstrate that EF-G variants are impaired in facilitating translation on AGA-bound ribosomes.”

- Ln 327-328- experiments assess relative error cluster formation, not rate of translation speed.

Reply: This is correct, thanks for pointing that out. Error cluster formation and its distance dependence provide indirect information on the residual speed of AGA-bound ribosomes, but do not allow a direct determination of translation rates as long as off rates for AGAs are not established. However, as we write in l. 244-247, “..., selective silencing of AGA-bound ribosomes in *fusA* mutant strains increases the chance that AGA will dissociate from the ribosome before several elongation cycles are completed, thereby suppressing error cluster formation.” We have revised the following sentence to read (l. 247-249): “**To assess whether translation proceeds slow enough for AGA to dissociate before multiple elongation cycles are completed**, we measured error cluster formation in wt and mutant strains in vivo using targeted MS. “

- p.11- Are the off rates of AGs bound to the translating ribosome known? If yes, is the model compatible with those rates?

Reply: To the best of our knowledge, the dissociation rate constants of AGAs from the ribosome have not been determined. While MICs, IC₅₀ and K_d values are well established, no quantitative assay currently exists that directly yields k_{off} values. Single-molecule-rinse-out experiments indicate that dissociation occurs at the time scale of seconds (Wasserman et al., 2015), which is qualitatively consistent with our model. For more quantitative comparisons, the k_{off} rates as well as the translation rates of AGA-bound ribosomes at low AGA concentrations (at high conc. often multiple binding sites are occupied) would need to be established, which is beyond the scope of this work.

- Would expression of wt EF-G overcome AGA resistance of the strains with mutations in chromosomal *fusA*?

Reply: Yes, there are examples in the literature showing that the resistance was reverted by complementing the *fusA* mutant strain with wild-type EF-G. We cite Ibacache-Quiroga et al. (2018) and Maunders et al. (2020) as examples in lines 92–93.

Minor points:

- Revise the sentence on lines 71-73: “...mutations...can confer high-level resistance, their impact on... resistance is limited...”

Reply: Thank you, we edited the sentence, and it now reads: “**Although 16S rRNA mutations in the ribosomal decoding center can confer high-level resistance, their impact is limited because most pathogens have multiple rRNA-encoding genes, so acquiring mutations in all copies simultaneously is unlikely**”.

- ref. to Supplementary Fig. 2 in lanes 94-96 is misplaced.

Reply: Changed as requested; the sentence now refers to Figure 1 and reads as follows: “Despite their clinical relevance, the mechanism by which *fusA* mutations confer resistance remains unclear, as EF-G cannot directly block AGA binding or contact the decoding center (Fig. 1A).”

- Lane 333, delete “of the”.

Reply: Done

-Lane 371, “than” not “that”.

Reply: Done

- Ln 436, correct “First, we first...”

Reply: Done

Reviewer #3 (Remarks to the Author):

The manuscript by Dastidar et al. reports the results of an original integrative study poised at unraveling a new antibiotic resistance mechanism, which neutralizes the action of several aminoglycosides (AGAs), through certain mutations in the EF-G gene, *fusA*. The authors show that EF-G resistance variants slow ribosome movement along mRNA when aminoglycosides are bound in a selective fashion. The delay resulting from the slowing down increases the chance that the drug dissociates before misreading occurs.

The authors found that *fusA* mutations confer resistance early in treatment by preventing self-promoted aminoglycoside uptake, which occurs consequently to the accumulation of misreading errors on the translated membrane proteins/transporters.

Indeed, the accumulation of faulty membrane proteins and misreading-induced metabolic by-products makes the inner membrane permeable to AGAs. As more and more AGAs enter the cell, more ribosomes bind AGAs and become corrupted. Over time, this self-promoted uptake leads to a massive influx of AGAs, resulting in a burst of translation errors leading ultimately to the damage of macromolecular structures, the disruption of the general metabolism and the membrane voltage dysregulation, all leading to cell death.

AGA resistance has been greatly characterized over the past decades and most AGA resistance mutations occur outside of the ribosome itself. Unexpectedly, a major hotspot of AGA resistance mutations is the *fusA* gene. Interestingly, *fusA* mutations confer resistance in various bacteria as well as in parasites such as *Leishmania*.

Despite their clinical relevance, the mechanism by which *fusA* mutations confer resistance remained thus far unclear, as EF-G neither blocks AGA binding nor contacts the decoding center.

The authors successfully demonstrate the existence of this previously unrecognized resistance mechanism that selectively silences corrupted targets, by applying a combination of quantitative mass spectrometry (MS), live-cell imaging, kinetic analysis, and cryo-EM, thus revealing a novel AGAs resistance strategy with potential therapeutic implications.

The authors found that *fusA* mutations specifically counteract the effects of certain AGAs that both disrupt translocation and induce misreading with error cluster formation.

The authors focused on three mutations in domain IV of EF-G (F593L, A608E and P610L). First, Dastidar et al. show that the proteomic changes induced by these mutations were minimal. They have analyzed the in-vitro translocation activity of the purified EF-G variants in the absence of AGAs and show that translocation rates substantially decreased in the presence of AGAs, supporting the idea that these mutants are selectively hindered in promoting translocation on AGA-bound ribosomes

The authors went on and probed the structure of an EF-G variant (P610L) in complex with the ribosome and Apramycin (Apr), using single-particle cryo-EM. The resulting structure superimposes nearly perfectly with EF-G wt and shows no significant structural changes in the decoding center or the drug-binding site, supporting the notion that EF-G variants do not displace the drug from the ribosome. They also show that the early stages of translocation up to GTP hydrolysis are unaffected by Apr but the later steps appear to be inhibited, particularly Pi release and the tRNA movement into chimeric states.

In summary, the authors show that *fusA* mutations silence AGA-bound ribosomes, preserving membrane integrity, and attenuating AGA uptake. This strategy minimizes translation errors without modifying the ribosome itself.

The manuscript is well written and the figures are clear. The references are sufficient, as far as the reviewer could tell and the performed experiments are state of the art. Furthermore, the depth of the performed analysis is simply overwhelming.

The reviewer has no major points to raise and only has very few minor comments. The manuscript can be in principle accepted in its current state.

Reply: We thank the reviewer's encouraging remarks and recognition of the contribution of our study.

Minor comments

Line 168 "these adaptations are likely due to changes in translation rates"

AdaptATions, not adaptations

Reply: Thank you for catching this. Done

Why was SlyD used as an in-vitro translation model? Why not other mRNAs?

Reply: We chose *slyD* for two reasons. First, the *in-vitro* translation of *slyD* mRNA is rapid and produces few translation intermediates. This is advantageous, as such intermediates could mask the effects of the EF-G variants under investigation. Second, although less important, SlyD is histidine-rich, which facilitates purification by nickel affinity chromatography if questions about its in-vivo relevance arise. Beyond these points, *slyD* simply serves as one representative model mRNA.

Figure 3 a-c:

Could the lack of significant changes in Apr binding on EF-G associated ribosomes, between the wt and the EF-G variant/mutant, rather be a consequence of the relatively modest resolution of the cryo-EM? Indeed, at 3Å many details can't be interpreted, such as solvent molecules and other subtle conformational changes with certainty...

Also, why is the resolution relatively modest at a time where structures of bacterial ribosomes are frequently below 2Å?

Reply: The present resolution of the key states is largely explained by the transient and dynamic nature of ribosome-EF-G complexes investigated here, while ribosome structures below 2Å usually describe more stable ribosomal states. Our structures show that EF-G mutation changes neither ribosome occupancy with EF-G or AGA, EF-G position, or the AGA interaction network on the ribosome, indicating that potential subtle unresolved differences do not have a significant impact on Apr binding to the ribosome.

Why was Apramycin used or considered as the archetype AGA?

Reply: Apramycin was chosen because it is a promising AGA candidate with markedly reduced ototoxicity compared to currently used AGAs. Additionally, its structure allows to evade many widespread resistance mechanisms. Therefore, it is important to investigate whether established resistance mutations also affect future drug candidates. Furthermore, apramycin is a good representative of its drug class. It shares ring I and II with DOS AGAs, binds to the canonical site in the decoding center, inhibits translocation, induces misreading and error clusters, and is subject to similar cellular resistance mutations.

From a strictly structural perspective, apramycin, with its 4-monosubstituted core, is not the archetype AGA. However, the concept of an “archetype” AGA is heuristic and an archetype AGA is difficult to define, as AGAs can be categorized by structural features, their kinetics on the ribosome, their cellular action, uptake, side effects, or resistance profiles. Given this diversity, we validated key aspects of the proposed mechanism for different AGAs to ensure generality.